# communications
# engineering

## REVIEW ARTICLE

# Spike frequency adaptation: bridging neural models and neuromorphic applications

Chittotosh Ganguly [1,6], Sai Sukruth Bezugam [2,6], Elisabeth Abs [3],
Melika Payvand [3], Sounak Dey[4] & Manan Suri [5 ✉]

The human brain's unparalleled efficiency in executing complex cognitive tasks stems from neurons communicating via short, intermittent bursts or spikes. This has inspired Spiking Neural Networks (SNNs), now incorporating neuron models with spike frequency adaptation (SFA). SFA adjusts these spikes' frequency based on recent neuronal activity, much like an athlete's varying sprint speed. SNNs with SFA demonstrate improved computational performance and energy efficiency. This review examines various adaptive neuron models in computational neuroscience, highlighting their relevance in artificial intelligence and hardware integration. It also discusses the challenges and potential of these models in driving the development of energy-efficient neuromorphic systems.

Spiking neural networks (SNNs) are inspired by their biological counterparts in which information is transmitted mostly through all-or-none events called spikes. Owing to the co-location of memory and computation within a spiking neuron, event-based asynchronous data processing, and sparse activations of nodes across the network, SNNs are inherently more power efficient compared to traditional deep neural networks that use continuous valued activation functions for the neurons[1–3]. SNNs, often mentioned as the third-generation artificial neural networks[4], are particularly suitable for temporal feature extraction and learning, as well as faster convergence to solutions for optimization problems[5,6]. Based on factors such as application requirement, computational complexity, and ease of implementation, different spiking neuron models are used in SNNs such as the Hodgkin–Huxley model, leaky-integrate and fire model, Izhikevich model and spike response model[7–14]. However, integrate and fire models[12] which mimic the activities of a biological neuron via functionalities of a simple resistance–capacitance electrical circuit are very popular due to their simple and elegant mathematical structure. An enhanced version of the integrate and fire model is the leaky integrate and fire (LIF) model which also takes the membrane voltage leak into account. SFA, i.e. increase in the inter-spike interval (ISI) over time for a regular spike train, is an intrinsic feature of biological neurons. In this paper, we will focus on SFA as an important feature to explore in SNNs.

In recent SNN models, adaptive neurons have been used to process temporal signals[15–19]. A recurrent spiking neural network (RSNN) aided with neurons with SFA is investigated in ref. [15], and is termed a long short-term memory neural network (LSNN). The addition of adaptive neurons improved the neural network's computational efficiency, compared to typical training using backpropagation through time (BPTT). The authors achieved an accuracy of 93.7% on sequential MNIST (SMNIST) and 66.7% on speech recognition (TIMIT data set). It has been demonstrated that the computational efficiency of RSNN has approached that of traditional long–short-term memory networks with neurons capable of SFA. The authors show that sparsely

[1] G. S. Sanyal School of Telecommunications, Indian Institute of Technology Kharagpur, Kharagpur, India. [2] Department of Electrical and Computer Engineering, University of California, Santa Barbara, USA. [3] Institute of Neuroinformatics, University of Zurich and ETH Zurich, Zurich, Switzerland. [4] TCS Research, Kolkata, India. [5] Department of Electrical Engineering, Indian Institute of Technology Delhi, Delhi, India. [6]These authors contributed equally: Chittotosh Ganguly, Sai Sukruth Bezugam. ✉email: manansuri@ee.iitd.ac.in

connected RSNNs with sparse firing can achieve all the above-mentioned tasks. The network can accomplish them due to the control of spike timings by SFA. In refs. [17,19], the authors enhanced the temporal computing capabilities of SNN enabled through SFA. In ref. [17], a single exponential model with two adaptation parameters has been used, while in ref. [19], a double exponential model with four parameters has been used for the same working memory task. It has been observed that for a working memory of 1200 ms, a double exponential model with high SFA converges much faster.

In[18] authors have utilized SFA for the development of a computational neurobiological model of language. For language processing short-term storage and integration of information in working memory is necessary. In the study, the authors offer a paradigm in which memory is sustained by intrinsic plasticity, which modulates spike rates. It has been shown that adaptive alterations via SFA produce memory on timescales ranging from milliseconds to seconds. The data is kept in adaptive conductances, which reduce firing rates and can be retrieved directly without the need for storage-based retrieval. Memory span is systematically connected to the adaptation time constant and baseline neuronal excitability levels. When adaptation is long-lasting, interference effects within memory develop.

Over the last few years, there has been an exponential increase in the research on adaptive neuron models. Though the major research is done in the neuroscience domain, it is slowly gaining momentum also in the domain of electronics and computer science, thanks to its increasing potential use in AI-based applications (Fig. 1). In this survey, we discuss the different adaptive neuron models and the corresponding SNN frameworks where these models have been employed. The adaptive neuron models from the computational neuroscience domain are described and the use of these models in engineering applications is highlighted. The remainder of the paper is organized as follows: Reasons for using adaptive neuron models are presented in the section "Why an adaptive neuron model". A detailed discussion of the available adaptive neuron models in the computational neuroscience literature is provided in the section "Description of adaptive neuron models". Section "State-of-the-art case studies with ALIF in SNN" considers selected applications that have been carried out employing adaptive neuron models. Section

"Hardware implementations of Adaptive neurons" presents hardware implementations of adaptive neuron models. The open challenges, road ahead, and future opportunities are presented in the section "Discussion and road ahead".

## Why an adaptive neuron model
Understanding and implementing SFA—which is observed widely in the biological neurons—in both computational models and hardware, could be leveraged to get a step closer to making artificial neural network computations more (power-) efficient. Here we first explain the biological phenomenon of SFA followed by its potential advantages in biology and for artificial intelligence.

**The biological phenomenon of spike frequency adaptation**. In biology, if a neuron is stimulated in a repeated and prolonged fashion, for example by constant sensory stimulation or artificially by applying an electric current, it first shows a strong onset response, followed by an increase in the time between spikes. Hence the spike rate attenuates and the so-called spike frequency adaptation takes place. Experimental data from the Allen Institute show that[17] a substantial fraction of excitatory neurons of the neocortex, ranging from 20% in the mouse visual cortex to 40% in the human frontal lobe, exhibit SFA as shown in Fig. 2a, b. There can be different causes for SFA: *First*, short-term depression of the synapse through depletion of the synaptic vesicle pool. This means that at the connection site between neurons, the signal from the pre-synaptic neuron cannot be transmitted to the next neuron. *Second*, by an increase in the spiking threshold of the post-synaptic neuron due to the activation of potassium channels by calcium, which has a subtractive effect on the input current. Hence, the same input current that previously caused a spike does not lead to a spike anymore. *Third*, lateral and feedback inhibition in the local network reduces the effect of excitatory inputs in a delayed fashion[20]. Therefore, like in the second case, spike generation is hampered.

Based on the biological description a large variety of spiking neuron models has been proposed in the literature, which implement SFA in different ways.

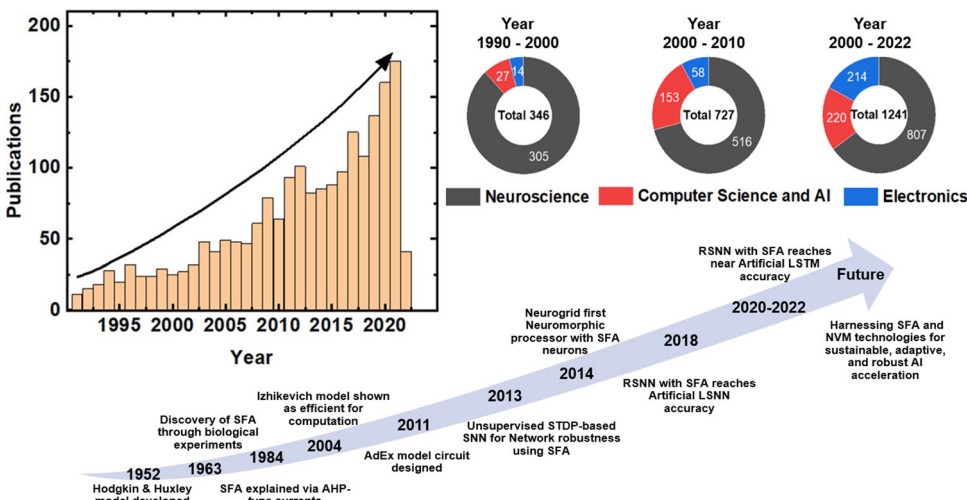

**Fig. 1 Growth and evolution of SFA research over the years.** Research on SFA has evolved significantly, expanding into diverse fields. The data, gathered from the Web of Science using keywords like "Spike rate adaptation", "Adaptive threshold" AND "neuron", and "Spike frequency adaptation" from 1990 to 2022, highlights growth in SFA research, initially rooted in neuroscience but now also prominent in computer science and electronics hardware. The milestones chart depicts key developments, the bar diagram represents total publications over time, and the pie charts break down the research focus across different periods. Future directions include harnessing SFA and emerging technologies for sustainable and innovative AI applications. AHP after hyperpolarization, STDP spike-timing-dependent plasticity

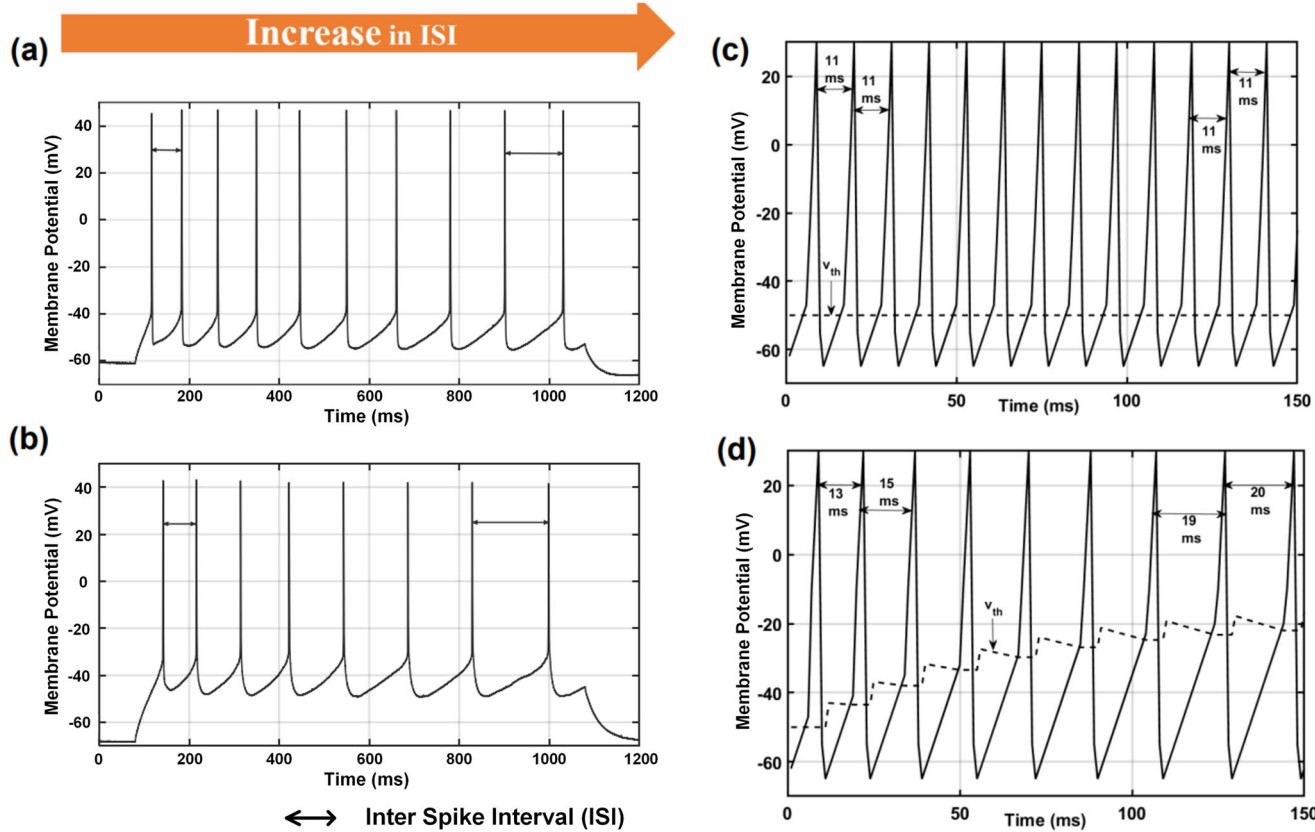

**Fig. 2 Adaptive LIF mimics SFA observed in biological neurons.** Biological neurons show SFA after prolonged stimulation. The response of two sample neurons from the Allen brain cell database[38] against a 1 s long step current is shown. Both neurons show SFA seen as an increase in the ISI. **a** Neuron from human temporal lobe (cell 601950719, sweep 44); **b** Neuron from mouse visual cortex (cell 595511209, sweep 33). **c** LIF, and **d** Adaptive LIF receiving an input Poisson spike train for 150 ms. Plot of membrane potential, $u(t)$, and corresponding threshold potential, $v_{th}$ are shown. The equations for LIF and ALIF are Eqs. (2) and (5). Parameters used in the adaptive threshold model shown (Eq. (5)): $v_{th}^{0} = 50$ mV, $\theta_{0} = 7$ mV, and $\tau_{\theta} = 100$ ms. LIF leaky integrate and fire, ALIF adaptive threshold leaky integrate and fire.

**Advantages of spike frequency adaptation.** From a biological standpoint, multiple advantages of the SFA mechanism have been observed. First, it lowers the metabolic costs, by facilitating sparse coding[21]: When there is no significant information in the presented inputs, as the input is either being repeated or there is a high-intensity constant stimulant, the firing rate is decreased leading to a reduction in metabolic cost and hence power consumption. Moreover, the separation of high-frequency signals from noisy environments is facilitated by SFA[22]. In addition, SFA can be seen as a simple form of short-term memory on the single-cell level[23].

In other words, SFA improves the efficiency[24] and accuracy of the neural code and hence optimizes information transmission[25]. SFA can be seen as an adaptation of the spike output range to the statistical range of the environment, meaning that it contrasts fluctuations of the input rather than its absolute intensity[26]. Thereby noise is reduced and, as mentioned above, repetitive information is suppressed which leads to an increase in entropy. Consequently, the detection of a salient stimulus can be enhanced[27]. These biological advantages of SFA can also be exploited for low-power and high-entropy computations in artificial neural networks.

To introduce SFA in spiking neural networks, a neuron model can be used which includes an adaptive threshold property[28]. SSNs with these kinds of neurons learn quickly, even without synaptic plasticity[29]. Moreover, SFA helps in attaining higher computational efficiency in SNNs[17]. For example, to achieve a store-and-recall cycle (working memory) of duration 1200 ms, a single exponential adaptive model requires a decay constant, $\tau_{a} = 1200$ ms in ref. [17], while a double exponential adaptive threshold model requires decay constants of $\tau_{a1} = 30$ ms and $\tau_{a2} = 300$ ms[19]—the latter being more efficient and sophisticated with four adaptation parameters compared to two parameters in ref. [17].

A comparison between baseline LIF behavior vs. SFA behavior through an adaptive LIF model is shown in Fig. 2c, d. In order to mimic constant current stimulation in the spiking domain, a high-frequency Poisson spike train ($f = 1000$ Hz of 150 ms duration, i.e. a spike is available at every time-step, $dt$) is applied to both LIF and adaptive LIF models. It can be observed that the LIF model produces 14 spikes, compared to 9 spikes for the adaptive model in the observed 150 ms time bin, leading to less spike handling operation in subsequent network layers. Moreover, the LIF model generates a spike train at a constant ISI of 11 ms, whereas SFA is observed in the spike train generated from the ALIF model. An ISI of 13 ms is observed for the first spike, and a non-decreasing ISI is observed further. Continuous adaptation of threshold voltage with every output spike for the ALIF model compared to the fixed threshold voltage for the LIF model leads to this SFA behavior.

In refs. [17,19,17], authors showed that SFA is also crucial for computation through spiking neurons. This function is particularly instrumental in overcoming the vanishing gradient problem in liquid state machines and RSNNs through the employment of an adaptive threshold, which serves as the source of SFA, within the gradient calculation process. Furthermore, the studies

provided evidence that SNNs equipped with SFA neurons are capable of achieving accuracy levels comparable to those found in artificial neural networks using long short-term memory networks.

Furthermore, the memory bottleneck in neural computation must be carefully considered, as memory access often consumes more time than computation itself, according to Wulf and McKee[30]. Within this context, the role of SFA becomes pertinent. Due to SFA, there is a decrease in spike frequency, leading to a corresponding reduction in the number of synaptic memory accesses, which are contingent on a pre-synaptic spike from the preceding layer. When compared to LIF models, this reduction in spikes has the potential to decrease computational efficiency. Further studies, as referenced in refs. [19,31,32], demonstrates that the benefits of SFA allow for a reduction in the number of neurons required to achieve similar accuracy in various spatio-temporal tasks. This reduction contributes to a decrease in both the area and energy footprint for the corresponding application.

## Description of adaptive neuron models

In this review paper, we have considered adaptive models based on the premise of the LIF framework. LIF models are popular in SNNs due to their simplicity, computational efficiency, and ability to capture some essential aspects of temporal character. Their simplicity makes LIF models amenable to theoretical analysis, which enables studying fundamental properties of SNNs, such as stability, dynamics, and network analysis. It is important to note that LIF models have their advantages but are still simplified abstractions of real biological neurons. An essential feature of a neuron missing in LIF is *spike frequency adaptation.*

Adaptive LIF models encompass all benefits of a LIF model and use a dynamic threshold that changes based on the neuron's recent activity. This mechanism can lead to more sophisticated information processing, as the neuron's sensitivity to input can be modulated by its recent firing history. With a more complex adaptation mechanism, the model attains higher efficiency with less iteration[16,19,33]. ALIF can replicate the phenomenon of SFA, where neurons become less responsive to repeated input spikes over time. This feature allows SNNs to capture more nuanced response patterns and better represent certain types of neural processing, increasing the computational efficiency as proven in the paper[16,19,33]. ALIF models can be easily combined with synaptic plasticity rules to study learning and memory processes in SNN. The adaptive behavior of these models allows for a more realistic exploration of synaptic strength changes and their impact on network function. ALIF models can also be implemented on neuromorphic hardware platforms, taking advantage of their more biologically plausible nature.

**Leaky integrate and fire model**. As already mentioned, LIF models are popularly used to mimic the spiking behavior of a neuron. The evolution of membrane potential, $u(t)$ in an LIF model can be written as[12,34]

$$\tau \frac{du}{dt} = RI(t) - [u(t) - v_{\text{rest}}] \tag{1}$$

where $\tau$ is the "leaky" time constant of the membrane, $R$ is the membrane resistance, $I(t)$ is the injected current, and $v_{\text{rest}}$ is the resting potential of the cell. In discrete time for spiking input Eq. (1) may be written as

$$u(t) = v_{\text{rest}} + RI(t) + (u_{t-1} - (v_{\text{rest}} + RI(t)))e^{\frac{-dt}{\tau}} \tag{2}$$

In SNN applications, $R$ is assumed to be unity and

$$I(t) = \sum_i w_i x_i(t) = \sum_i w_i \delta_i(t) \tag{3}$$

where $w_i$ is the synaptic weight between target neuron and $i$th pre-synaptic neuron and $x_i(\delta_i)$ corresponding spiking input to the $i$th pre-synaptic neuron. When membrane potential, $u(t)$ at $t = t^{(f)}$ crosses a predefined fixed threshold, $v_{\text{th}}^0$, a spike is generated i.e.

$$t = t^{(f)} \iff u(t = t^{(f)}) = v_{\text{th}}^0 \tag{4}$$

**Adaptive LIF**. In adaptive LIF, a time-dependent function $\theta(t)$ is added to the fixed threshold, $v_{\text{th}}^0$ after every spike causing an adaptation of the threshold. The threshold potential, $v_{\text{th}}(t)$, gradually returns to its steady state value depending on threshold adaptation time constant $\tau_\theta$. The expression for adaptive threshold is thus given as[12]

$$v_{\text{th}}(t) = v_{\text{th}}^0 + \sum_f \theta(t - t^{(f)}) = v_{\text{th}}^0 + \sum_f \theta_0 \exp[\frac{-(t - t^{(f)})}{\tau_\theta}] \tag{5}$$

where the function $\theta(t)$ is

$$\theta(t) = \theta_0 \exp[-t/\tau_\theta] \tag{6}$$

when membrane potential, $u(t)$ reaches a threshold, it is reset to $v_{\text{rest}}$

$$u(t) \geq v_{\text{th}}(t) \implies v(t) = v_{\text{rest}} \tag{7}$$

**Double EXponential Adaptive Threshold (DEXAT)**. A Double EXponential Adaptive Threshold (DEXAT) neuron model has been proposed by Shaban et al.[19]. The authors demonstrated that the proposed DEXAT model provides higher accuracy, faster convergence, and flexible long short-term memory (working memory in neuroscience terms) compared to existing counter parts in the literature.

The membrane potential dynamics are described through Eq. (1). The threshold adaptation rule is given by the following set of equations:

$$v_{th_j}(t) = b_{j0} + \beta_1 b_{j1}(t) + \beta_2 b_{j2}(t) \tag{8}$$

$$b_{j1}(t + \delta t) = \rho_{j1} b_{j1}(t) + (1 - \rho_{j1})z_j(t)\delta(t) \tag{9}$$

$$b_{j2}(t + \delta t) = \rho_{j2} b_{j2}(t) + (1 - \rho_{j2})z_j(t)\delta(t) \tag{10}$$

where $\rho_{j1} = \exp[\frac{-\delta t}{\tau_{b_1}}]$ and $\rho_{j2} = \exp[\frac{-\delta t}{\tau_{b_2}}]$ control the evolution of adaptive threshold with time, where $\tau_{b_1}$ and $\tau_{b_2}$ are threshold adaptation time constants and $\beta_1$ and $\beta_2$ are two scaling factors ($\beta_1, \beta_2 > 0$). For each spike $z_j(t)$, threshold potential $v_{\text{th}_j}(t)$ increases by $\frac{\beta_1}{\tau_{b_1}} + \frac{\beta_2}{\tau_{b_2}}$.

**Multi-time scale adaptive threshold**. In this model, the behavior of the membrane potential is governed by Eq. (1) as well. The threshold potential is also increased from its present value whenever a spike is generated. The threshold gradually decays to the resting potential, $v_{\text{rest}}$ depending on the decay time constants. The rule for threshold update[35] is given below:

$$v_{\text{th}}(t) = \sum_i H(t - t_i) + v_{\text{rest}} \tag{11}$$

where $t_i$ is the $i$th spike time. The form of $H(t)$ is described as

$$H(t) = \sum_{j=1}^{L} \alpha_j \exp[\frac{-t}{\tau_j}] \tag{12}$$

where $L$ is the number of threshold time constants, $\tau_j$ is $j$th time constant ($j = 1, 2, ..., L$) and $\alpha_j$ is the weight of the $j$th time constant.

**Adaptive Exponential (AdEx) LIF**. Adaptive exponential LIF model involves two state parameters, membrane potential, $u(t)$ and adaptation variables $w_k$ to explain various spiking dynamics[12,34]. The evolution of $u(t)$ and $w_k$ are described by the following equations:

$$\tau_m \frac{du}{dt} = f(u) - R\sum w_k + RI(t) \tag{13}$$

$$\tau_k \frac{dw_k}{dt} = a_k(u - v_{rest}) - w_k + b_k \tau_k \sum_{t^{(f)}} \delta(t - t^{(f)}) \tag{14}$$

A popular choice of $f(u)$ is mentioned in ref. [12]

$$f(u) = -(u - v_{rest}) + \Delta_T \exp\left(\frac{u - v_{th}}{\Delta_T}\right) \tag{15}$$

where $\Delta_T$ is the sharpness parameter, $v_{th}$ threshold potential, $w_k$ adaptation current, $a_k$ adaptation parameter, $b_k$ amount by which adaptation current increases after threshold.

**Spike response model**. The spike response model (SRM) is a generalization of the leaky integrate-and-fire model[12]. In contrast to the LIF model, SRM includes refractoriness behavior in the model equation itself. While the membrane potential of an integrate-and-fire model is described using coupled differential equations, SRM is formulated using filters.

The membrane potential, $u(t)$, in the presence of an external current, $I(t)$, is given below as mentioned in refs. [12,36]

$$u(t) = \sum_{f \in F} \eta(t - t^{(f)}) + \int_0^\infty k(s)I(t - s)ds + v_{rest} \tag{16}$$

Here, the function, $k(t)$, describes the filter of the voltage response to a current pulse. Input current $I(t)$ is filtered with a filter $k(t)$ and produces corresponding input potential $h(t) = \int_0^\infty k(s)I(t - s)ds$. A spike occurs when the membrane potential, $u(t)$, reaches the threshold $v_{th}(t)$. The membrane potential after a spike is described by a function $\eta(t)$. The function, $\eta(t)$ models the refractory behavior after a spike. The set $F$ is a collection of all spike times before $t$ and is defined as

$$F = \{t^{(f)}; 1 \le f \le n : u(t^{(f)}) = v_{th}(t^{(f)})\} \tag{17}$$

The threshold for a spike generation in SRM is not fixed and is time-dependent, denoted by $v_{th}(t)$. A spike is generated when the membrane potential, $u(t)$ crosses the dynamic threshold $v_{th}(t)$. The expression of spike time $t^{(f)}$ is given as

$$t = t^{(f)} \iff u(t) = v_{th}(t)|_{t=t^{(f)}},$$
$$\frac{d[u(t) - v_{th}(t)]}{dt} \ge 0|_{t=t^{(f)}} \tag{18}$$

A standard model of the dynamic threshold is

$$v_{th}(t) = v_{th}^0 + \sum_{f \in F} \theta(t - t^{(f)})$$
$$= v_{th}^0 + \int_0^\infty \theta(s)S(t - s)ds \tag{19}$$

Here, $v_{th}^0$ is the threshold in the absence of spiking for a long duration. The threshold potential is increased by the function $\theta(t)$ after each output spike for $t^{(f)} < t$.

In SRM, when the input is a spike train, the equation for membrane potential $u(t)$ is modified as:

$$u(t) = \sum_{f \in F} \eta(t - t^{(f)}) + \sum_j \sum_{g \in F_j} w_j \varepsilon(t - t_j^{(g)}) + v_{rest} \tag{20}$$

where $w_j$ is the weight of the synapse connected to the target post-synaptic neuron through $j$th pre-synaptic neuron. $F_j$ is the set of all spike times of $j$th pre-synaptic neuron. The spike time of $g_{th}$

spike from $j$th pre-synaptic neuron is denoted by $t_j^{(g)}$. The function $\varepsilon(t)$ denotes spike response function.

**Generalized LIF (GLIF)**. Researchers of the Allen Institute for Brain Science proposed five Generalized Leaky Integrate and Fire (GLIF) models by updating the baseline LIF model[37]. Three primary factors that have been considered while updating the baseline LIF model are: (i) membrane and threshold potential reset rule after a spike, (ii) slow affecting current from $Na^+$ and $K^+$ channels which have been activated during a spiking phenomenon, (iii) changes in threshold potential caused by sub-threshold potential and spikes[38]. Five GLIF models are found in the literature, namely GLIF-I to GLIF-V. The details of the five GLIF models are as follows:

*GLIF-I*. Basic LIF model as described in the section "Leaky integrate and fire model ".

*GLIF-II*. GLIF-II incorporates biologically reset rule on top of the GLIF-I. The equation for spike-induced threshold is

$$\frac{d\theta_s}{dt} = -b_s \theta_s \tag{21}$$

When membrane potential $u(t) \ge v_{th} + \theta_s$, it resets to

$$u(t_+) = v_{rest} + f_v(u(t_-) - v_{rest}) - \delta u$$
$$\theta_s(t_+) = \theta_s(t_-) + \delta\theta_s \tag{22}$$

where $f_v$ is the multiplicative coefficient and a threshold component $\delta\theta_s$ has been added after every spike to $\theta_s(t)$.

*GLIF-III*. Slow fluctuating currents for the activated $Na^+$ and $K^+$ ion channels for a spike have been included in GLIF-III. These current components are modeled below as described in refs. [37,39]:

$$\frac{dI_j(t)}{dt} = -k_j I_j(t), j = 1, 2, \dots, N \tag{23}$$

Like GLIF II, if $u(t) \ge v_{th}$, the membrane potential $u(t)$ is reset to $v_r$ and current components $I_j(t)$ are updated as

$$I_j(t_+) = R_j I_j(t_-) + A_j \tag{24}$$

where $k_j$, $R_j$, and $A_j$ are post-spike current time constant, a multiplicative constant (typically $R_j = 1$) and after-spike current amplitude, respectively.

*GLIF-IV*. It combines both GLIF-II and GLIF-III models. It has both biologically defined reset, after spike current components and a spike induced threshold potential[37,39].

*GLIF-V*. Along with after-spike currents $I_j(t)-$s, and spike-induced threshold component $\theta_s(t)$, a sub-threshold potential-induced threshold variable $\theta_u(t)$ is defined in GLIF-V. The model has four state parameters viz. $u(t)$, $I_j(t)$, $\theta_s(t)$ and $\theta_v(t)$ [37,39]. When $u(t) \ge \theta_v + \theta_s$, a spike is generated and state variables are updated following the reset rule described below:

$$I_j(t_+) = R_j I_j(t_-) + A_j$$
$$u(t_+) = v_{rest} + f_v(u(t_-) - v_{rest}) - \delta u$$
$$\theta_s(t_+) = \theta_s(t_-) + \delta\theta_s$$
$$\theta_u(t_+) = \theta_u(t_-) \tag{25}$$

where $a$ and $b_u$ are adaptation index of sub-threshold potential dependent threshold component and sub-threshold potential-induced threshold time constant.

The computational complexity of the available neuron models reported in the literature is calculated in terms of the number of

**Table 1 Summary of a selection of adaptive neuron models based on computational complexity**

| Model | Membrane potential equation | Adaptive threshold equation | Number of arithmetic operations required per iteration |
|---|---|---|---|
| LIF | $\tau \frac{du}{dt} = RI(t) - [u(t) - v_{rest}]$ | $v_{th}^0$ fixed threshold | 10 |
| Adaptive LIF | $\tau \frac{du}{dt} = RI(t) - [u(t) - v_{rest}]$ | $v_{th}(t) = v_{th}^0 + \sum_f \theta(t - t^{(f)}) = v_{th}^0 + \sum_f \theta_0 \exp - \frac{(t - t^{(f)})}{\tau_\theta}$ | $6F + 10$ |
| DEXAT | $\tau \frac{du}{dt} = RI(t) - [u(t) - v_{rest}]$ | $v_{th_j}(t) = b_{j0} + \beta_1 b_{j1}(t) + \beta_2 b_{j2}(t)$ | 29 |
| Multi-scale adaptive threshold | $\tau \frac{du}{dt} = RI(t) - [u(t) - v_{rest}]$ | $v_{th}(t) = \sum_i H(t - t_i) + v_{rest}$ | $5LF + 10$ |
| SRM | $u(t) = \sum_{f\in F}\eta(t - t^{(f)}) + \int_0^\infty k(s)I(t - s)ds + v_{rest}$ | $u(t) = \sum_{f\in F}\eta(t - t^{(f)}) + \sum_j \sum_{g\in F_j} w_j \varepsilon(t - t_j^{(g)}) + v_{rest}$ | $2NF + 8F_pN$ |
| Adaptive Exponential Model | $\tau_m \frac{du}{dt} = f(u) - R\sum w_k + RI(t)$ | $\tau_k \frac{dw_k}{dt} = a_k(u - v_{rest}) - w_k + b_k \tau_k \sum_{t^{(f)}} \delta(t - t^{(f)})$ | $2N_{ad}F + 7N_{ad} + 13$ |
| GLIF I | $\tau \frac{du}{dt} = RI(t) - [u(t) - v_{rest}]$ | $v_{th}^0$ fixed threshold | 10 |
| GLIF II | $\tau \frac{du}{dt} = RI(t) - [u(t) - v_{rest}]$ | $\frac{d\theta_s}{dt} = -b_s \theta_s$ | 13 |
| GLIF III | $\tau \frac{du}{dt} = RI(t) - [u(t) - v_{rest}]$ | $\frac{dI_j(t)}{dt} = -k_j I_j(t), j = 1, 2, ..., N$ | $4N_{ad} + 10$ |
| GLIF IV | $\tau \frac{du}{dt} = RI(t) - [u(t) - v_{rest}]$ | $I_j(t_+) = R_j I_j(t_-) + A_j$ | $4N_{ad} + 14$ |
| GLIF V | $\tau \frac{du}{dt} = RI(t) - [u(t) - v_{rest}]$ | $I_j(t_+) = R_j I_j(t_-) + A_j$ | $4N_{ad} + 24$ |

$N$ is the number of pre-synaptic neurons connected to the target neuron, $L$ is the number of the exponential kernels used to approximate threshold potential, $N_{ad}$ is the number of adaptation variables used in the model equation, $F$ and $F_p$ are the number of spikes generated in the target neuron and connected pre-synaptic neuron respectively.

arithmetic operations (number of arithmetic additions and multiplications) required in an iteration. A summary of the computational complexity of the adaptive spiking neuron models is reported in Table 1.

While the number of arithmetic operations per iteration required is often used as a proxy for computational complexity, it's essential to recognize that it doesn't linearly correlate with power consumption. Energy efficiency depends on various factors, including hardware design, memory access patterns, and algorithmic optimizations.

**State-of-the-art case studies with ALIF in SNN**

In this section, we will discuss a selection of applications that use the aforementioned adaptive neuron models.

The GLIF-II model has been used in refs. [15–17] to implement STORE-RECALL, video recognition, image classification, delayed XOR, and cognitive computational tasks. The property of SFA through the ALIF model is exploited in the above works. On the Google speech data-set, delayed XOR, and cognitive computation task $12AX$, authors in ref. [17] have achieved an accuracy of $90.88 \pm 0.22\%$, $95.19 \pm 0.014\%$, and $92.89\%$ respectively. In ref. [15], an accuracy of 93.7% on SMNIST and 66.7% on speech recognition (TIMIT data-set) have been obtained. Bellec et al.[16] have performed a STORE-RECALL task of 1200 ms with a classification rate of 95% in 50 iterations.

The learning algorithm used in refs. [15,17] BPTT. A learning algorithm, called e-prop for RSNNs, which is an alternative to BPTT is proposed in ref. [16].

Multiple spatio-temporal applications were shown in[19] using DEXAT neuron model. One of the simplest benchmarks was done through STORE and RECALL task, where working memory is considered as the time gap between STORE and RECALL instructions. An LSNN consisting of 10 LIF and 10 DEXAT neurons was used for the task. The network was trained for 200 ms with a minimum desired decision error of 0.05. The results indicate that to achieve a working memory of 1200 ms, $\tau_{b1}$ and $\tau_{b2}$ need to be 30 and 300 ms, respectively. However, increasing $\tau_{b2}$ to 500 ms led to an even faster convergence of the LSNN network for the same working memory. Compared to the working memory value with the DEXAT model, these values of $\tau_{b1}$ and $\tau_{b2}$ are much smaller. However, in ref. [17] the value of single threshold adaptation time constant $\tau_b$ is comparable to

working memory, which is a clear disadvantage compared to the model[19].

A system-level simulation of LSNN with DEXAT reported classification accuracy of 96.1% on sequential MNIST (SMNIST) i.e. converging in 30% fewer epochs to a higher accuracy. Further, they evaluated a spatio-temporal voice recognition application using the Google Speech Command (GSC) dataset. They had achieved a 91% accuracy using a single hidden recurrent layer.

Using two hidden layers of GLIF-II and varying adaptation time values for each layer,[40] demonstrated an accuracy of 92.1% on the GSC data set. In addition, the study shows the usefulness of adaptive neurons for tasks with an inherent temporal dimension, such as the categorization of ECG wave patterns (accuracy 85.9%) and gesture recognition using a radar spectrogram.

Wade et al. [41] used a variant of adaptive LIF (Eq. (5)) for classification tasks. A supervised learning algorithm called *Synaptic Weight Association Training* (SWAT), a variant of STDP, is used here. It provides a classification accuracy of 95.3%, 96.7%, and 95.25% for Iris, Wisconsin Breast Cancer, and TI46 speech corpus data-sets, respectively. The membrane potential dynamics of the model used here are governed by Eq. (1).

An SNN-based computing paradigm has been proposed to provide immunity from device variations for memristive nano-devices in ref. [42]. The neuron model used in this paper is LIF in nature. A dynamic threshold is designed through homeostasis. The adaptive threshold and lateral inhibition help a specific group of neurons to respond to a particular stimulus[42]. The network is tested on the MNIST data-set. It achieves a maximum of 93.5% accuracy with 300 output neurons. A system-level simulation shows that the designed device can tolerate parameter variation up to 50% of the standard deviation of parameter values.

In[43], Diehl et al. created An SNN with an ALIF model for digit recognition on the MNIST benchmark. The model is a synaptic conductance-based LIF and an adaptive threshold has been implemented following Eq. (5). The average classification accuracies on the MNIST data-set of 82.9%, 87%, 91.9%, and 95% have been achieved by the model of 100, 400, 1600, and 6400 neurons, respectively.

Recently, Jiang et al.[44] demonstrated the use of adaptive neurons for arrhythmia detection on edge devices with a non-recurrent SNN. In[45], authors have shown the potential of adaptive neurons used on event-based sensor data for unsupervised optical flow

estimation. Encoding international morse code was demonstrated by adjusting the threshold of neurons adaptively in an SNN through reinforcement learning[46]. In ref. [17], authors have shown that SFA can help in efficient network computations for temporally dispersed data. Using the same neuron model in ref. [47] a sparse RSNN, based on ALIF was used successfully to extract relations between words and sentences in a text in order to answer questions about the text. Apart from the adaptive neuron models discussed in the section "Description of adaptive neuron models", a few additional adaptive neuron models have been explored in the literature. Details of those models and associated applications are highlighted below.

In ref. [48], author proposed an adaptive threshold module (ATM) for An SNN based architecture. ATM algorithm controls internal threshold potential. This ATM is used to control output firing rate, which helps to to extract the information encoded in input stimulus. The model is validated on speech TIDIGITS and RWCP data-sets.

The model is tested against Poisson spike trains for various frequencies and lengths, TIDIGITS Speech and RWCP data-sets. For a Poisson spike train of 300 Hz with 4000 patterns, ATM model with two-phase classifier shows an accuracy of 96.1%. The accuracy for TIDIGITS and RWCP data-sets are 99.5% and 97.64%, respectively.

Another variant of ALIF has been proposed in ref. [49] and has been implemented on feed-forward SNN using STDP. The model is validated through MNIST data-set. The maximum achieved classification accuracy with MNIST data-set is 82%.

The selected works presented in this section are summarized in Table- 2.

Further, a comparison of the neuron models listed in Table 2 in terms of flop counts is provided in Table 3.

*Observation:* Tables 1 and 3 illustrate that the number of arithmetic operations required to implement LIF and DEXAT models is not the same. In Table 1, an external current injection is assumed following the traditional approach for a single isolated neuron; whereas in Table 3, the numbers are reported when those isolated neurons are used together to implement a spiking network, where input current is described through Eq. (3).

In the next section, we discuss hardware implementations of adaptive neurons and highlight different simulators that support adaptive neurons.

## Hardware implementations of adaptive neurons

The integration of SFA models within hardware has progressively manifested as a seminal approach to augmenting the efficiency of AI hardware, with promising applications in neuromorphic computing. Existing *Commercial Off-the-Shelf* (COTS) platforms, deploying Leaky Integrate-and-Fire (LIF) neuron blocks as fundamental units, have seen several studies for implementing SFA neurons in multicompartment neuron configurations[31,50–52].

The recent developments in the field, such as the work by Bezugam et al.[31] have proven the feasibility of achieving resource utilization with reduced neuron count. Further, Intel's Loihi-2 architecture has ventured into adding ALIF models, heralding a promising avenue in *Non-Volatile Memory* (NVM) based hardware.

Parallel to these advancements, hardware implementations of neuron models such as Integrate and Fire (IF), LIF[53–55] and Adaptive Exponential LIF[56–58] have been widely reported, encompassing complementary metal oxide semiconductor (CMOS). Moreover, many designs have exploited emerging resistive memory technologies for such implementations, such as using RRAM[59], PCM[60], and CBRAM[61–67]. Recently, superconducting device, 2D material-based device neuron circuits had shown SFA[68,69].

Digital implementation of modified AdEx neuron models on FPGA further amplifies the possibilities[70–72]. Innovations such as[73] demonstrate improvements in speed and footprint without compromising neuronal dynamics. The utilization of quantized versions of DEXAT neuron models[19] represents another noteworthy advancement (see Fig. 3). Notably, the integration of SFA within FPGA has led to the development of a pre-synaptic spike-driven architecture, which significantly reduces resource utilization and buffer size for caching events, while maintaining accurate task-solving performance[74].

The confluence of these developments underlines the multidimensional potential of SFA within neuromorphic hardware. The exploration of digital circuits, analog designs, and emerging NVM devices presents a diverse spectrum of opportunities and challenges. The emergence of space-efficient and low-power circuits constructed with advanced 3D integration technologies indicates the path forward.

The adoption and adaptation of SFA within neuromorphic hardware demonstrate a forward-thinking approach in both design complexity and efficiency. This integration harbors significant potential not only in optimizing resource utilization but also in paving the way for future innovations. The collaborative intersection between various technologies and methodologies emphasizes the vibrant dynamism in this field. As evidenced by recent developments, the application of SFA in hardware is not a mere theoretical prospect but a tangible trajectory that stands to redefine the next generation of neuromorphic computing.

**Simulators supporting adaptive neuron models**. Various simulators support adaptive neuron models for building SNN. The function of the AdEx neuron model is based on polarizing and hyperpolarizing currents supported by PyNN[75], BRIAN2[76] and NEST[77]. Neko[78], FABLE[79] and Norse[80] are SNN simulation frameworks based on PyTorch that enable the ALIF neuron model for constructing Recurrent-SNN. Here, the ALIF neuron model is a state function in which the membrane voltage and neuron threshold are updated with every iteration. More hardware-realistic neuromorphic circuit simulation is shown in[72]. While this list encapsulates a range of simulators pivotal to SFA-based SNN simulation, it is imperative to note that the spiking neural network landscape is rich and continually expanding, with numerous other simulators also playing crucial roles in advancing this field.

Different factors, along with the adaptive neuron model, that play a vital role in accomplishing a particular task using SNN are highlighted in the next section.

## Discussion and road ahead

The preceding exploration of SFA has offered significant insights into its principles, neuron models, applications, and hardware implementations. As the field advances, the complexity and potential of SFA continue to unfold, demanding innovative approaches and broader research horizons.

In this section, as shown in Fig. 4 we identify remaining challenges (Fig. 4a) and delineate a roadmap for future research (Fig. 4b), building upon the scientific understanding cultivated herein. By synthesizing the current state of the field with a forward-looking perspective (Fig. 4c), we aim to contribute a decisive and thoughtful conclusion to the ongoing discourse on SFA.

## Challenges and roadmap

*Encoding techniques in SFA*. Traditional spike encoders such as Poisson[43], rate-based encoding[81], and population encoding[17], often struggle to capture the complex dynamics inherent to SFA, potentially leading to information loss[24,82]. In the context of biological systems, neural adaptation serves as a crucial tool for

**Table 2 Summary of the selected works using adaptive neuron models**

| Work | Neuron model | Adaptation equation | Learning rule | Performance |
|---|---|---|---|---|
| Wade et al. [41] | Adaptive LIF | $v_{th_{new}} = mv_{th_{orig}}\exp\left[\frac{t-t_i}{\tau_{decay_j}}\right]$ | BCM rule with STDP | 95.3% on Iris, 96.7% for Wisconsin data-sets, 95.25% for TI46 speech corpus |
| Querlioz et al. [42] | LIF and homeostasis | $\frac{dv_{th}(t)}{dt} = \gamma(A - T)$ | Customized STDP | 93.5% on MNIST dataset |
| Amin [48] | Modified SRM | $v_{th}(t_i) = \frac{\delta t}{t_i - t_{i-1}} + v_{th}(t_{i-1})$ | Local Learning Rule $w_i = \beta_1 t_i$ or $w_i = \frac{\beta_2}{t_i}$ | 96.1% Poisson spike train, 99.5% for TIDIGITS data set, 97.64% for RWCP data set |
| Liu et al. [49] | Adaptive LIF | $v_{th}(t) = k \times \max[u(t)]$ | STDP | 82% on MNIST data-set |
| Diehl et al. [43] | LIF and homeostasis | $v_{th}(t) = v_{th}^0 + \sum_f \theta_0 \exp\left[\frac{-(t-t^{(f)})}{\tau_{theta}}\right]$ | STDP with pre and post-synaptic traces | 95% on MNIST data-set |
| Salaj et al. [17] Bellec et al. [15,16] | Adaptive LIF | $A_j(t) = v_{th} + \beta a_j(t),$ $a_j(t+\delta t) = \rho_j a_j(t) + (1-\rho_j)z_j(t)\delta(t)$ | BPTT [15,17], E-PROP [16] | [17]90.88 ± 0.22% on Google speech data-set 95.19 ± 0.014% on Delayed XOR 92.89% on Cognitive computation task 12AX[15]93.7% on SMNIST 66.8% on speech recognition (TIMIT data-set)[16] STORE-RECALL task of 1200 ms with classification rate of 95% |
| Shaban et al. [19] | DEXAT | $v_{th_j}(t) = b_{j0} + \beta_1 b_{j1}(t) + \beta_2 b_{j2}(t)$ | BPTT | Working memory of 1200 ms, For $\tau_{b1} = 30$ ms and $\tau_{b2} = 300$ ms. 96.1% on SMNIST, 91% on GSC data set |
| Yin et al. [40] | Adaptive LIF | $\eta_t = \rho\eta_{t-1} + (1-\rho)S_{t-1}, v_{th} = b_0 + \beta\eta_t$ | BPPT, Surrogate-Gradient, and Multi-Gaussian | Multiple performances on various datasets |
| Vallés et al. [45] | LIF and homeostasis | $\lambda_X \frac{dX_{i,j,d}(t)}{dt} = -X_{i,j,d}(t) + \alpha S_j^{l-1}(t - t_d)$ | STDP | - |
| Rao et al. [47] | LIF and after hyper-polarizing currents | $i_{AHP_j}[t + \Delta t] = \alpha_{AHP} i_{AHP_j}[t] - \beta z_j[t]$ | Surrogate gradient (BPTT) | 96.09% on SMNIST |

### Table 3 A comparison of the neuron models in terms of computational complexity

| Model | Number of arithmetic operations required per Iteration |
|---|---|
| LIF | $2N + 6$ |
| Wade et al.[41] | $2N + 12$ |
| Querlioz et al.[42] | $2N + 11$ |
| Amin[48] | $4N + 4$ |
| Liu et al.[49] | $8NF + 12$ |
| Diehl et al.[43] | $6F + 22$ |
| Salaj et al.[17] Bellec et al.[15,16] | $2N + 17$ |
| Shaban et al.[19] | $2N + 25$ |
| Yin et al.[40] | $2N + 18$ |
| Vallés et al.[45] | $11NS - 1$ |
| Rao et al.[47] | $3N + 19$ |

*N is the number of connections to the target neuron, F is number of spikes produced by the target neuron in an observation interval, and S is the number of synapses between a connected pre-synaptic neuron and the target neuron.*

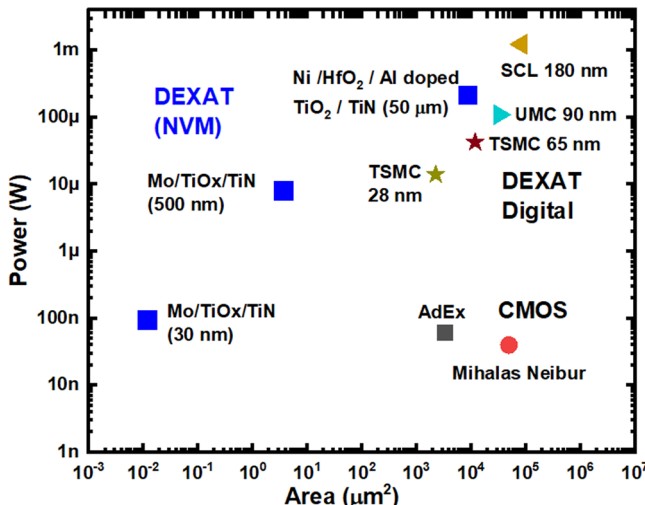

**Fig. 3 Power vs. area comparison for various realizations of adaptive neuron models.** DEXAT neuron threshold circuit (digital, NVM based)[19], AdEX[94], Mihalas-Niebur[95] analog CMOS adaptive neuron circuits. NVM non-volatile memory, SFA spike frequency adaptation.

calibrating sensitivity across diverse intensity gradients, illuminating the need for specialized SFA-based encoders designed to emulate these biological nuances[83]. The issue becomes more pronounced when translating these principles into hardware systems. Here, temporal sensitivity to presynaptic spikes is elevated within SFA-based neurons; a slight misalignment in spike timing can result in considerable information loss, unlike in non-SFA neurons, where pre-synaptic spike timing allows for greater flexibility. This dilemma necessitates an exploration of dynamical encoding schemes inspired by information theory, a venture that could substantially reduce information attrition. The scant existing research into the compatibility of these encoding techniques with SFA-based neural networks further emphasizes the urgent need for novel strategies. Such innovation will not only minimize the loss of information but also expand the practical applicability of SFA in encoding, presenting a significant frontier in neural computation.

*Learning algorithms and adaptive neurons.* The deployment of learning rules and adaptive neurons in SFA presents unique challenges, despite some recent advancements. The utilization of the pseudo-gradient by Salaj et al.[17] in BPTT and the online version of BPTT (Eprop)[33] represents a noteworthy stride towards accommodating SFA dynamics in learning. These developments incorporate adaptive thresholds but falter when the number of layers increases significantly. With an escalation in layers, there may be fewer spikes exhibiting SFA, rendering the employment of SFA-based pseudo-gradient costly and less effective. Furthermore, many properties intrinsic to SFA, such as temporal sensitivity, adaptation to stimulus statistics, and independent firing transitions, still await integration into learning algorithms. The ALIF[17], although aligned with certain biological properties, falls at the lower end of the biological realism spectrum. Exploration of other neuron models, with unique features complementing SFA, is needed. Additionally, higher-order spike response models (SRMs) could provide enhanced dynamics that may augment learning but require profound investigation. The search for learning rules aligned with SFAs complexities remains a challenge, necessitating innovative algorithms to optimize adaptive neuron functionality.

In the future, research in SFA-based neuron models could focus on how the different implementation options for SFA (intrinsic changes of the spiking threshold vs. inhibitory input vs. short-term synaptic plasticity), including their different time-scales, affect the coding properties of a network. Hence, investigate if some motifs are more suitable for certain computations than others. In addition,

it would be interesting to investigate how adaptation propagates across layers, which would help in understanding how SFA occurring in one brain region affects the computation in its downstream regions.

*Network architecture and connectivity.* Recent studies underscore the complexity and potential advantages of integrating diverse neuron types within an RSNN[84], reflecting the intricate interactions present in biological networks[17,19]. The introduction of sparsity into networks can lead to challenges with SFA-based neurons, as the heavily decreased input firing may conflict with the unique properties of these neurons. This raises both potential benefits and problems in terms of information processing and network efficiency. Consequently, the careful selection of the location of SFA neurons within the network becomes an essential criterion. Notably, the regularization of the firing rate of output neurons in unsupervised SNNs, such as through homeostasis as seen in the work of Diehl and Cook, highlights that even before SFA-based networks were prevalent, there were instances of support for multiple neuron types[42,43]. Exploration into graph-based Hopfield networks for combinatorial optimization offers a promising avenue, as evidenced by the recent demonstration of a thermal neuron exhibiting SFA behavior[85]. However, this field remains largely under-researched. The challenge, therefore, lies in systematically understanding and capitalizing on the unique dynamics of SFA, considering architecture design, optimization strategies, connectivity schemes, and the nuanced interplay with sparsity.

*Hyperparameter tuning and mathematical complexity of SFA models.* The hyperparameter tuning of SFA models poses a complex problem, demanding an intricate balance between biological formalism and computational efficiency. The grid-based search methods typically employed may fall short in such complex scenarios. An exploration of advanced optimization techniques, such as Bayesian optimization or gradient-based optimization, is suggested as a possible avenue for more intelligently and efficiently navigating the parameter space specific to SFA models. Systematic ablation studies could enhance this process by elucidating the effects of individual parameters and their interactions, potentially leading to a deeper understanding of hyperparameter significance. The computational cost of implementing adaptive neurons in SFA, especially when

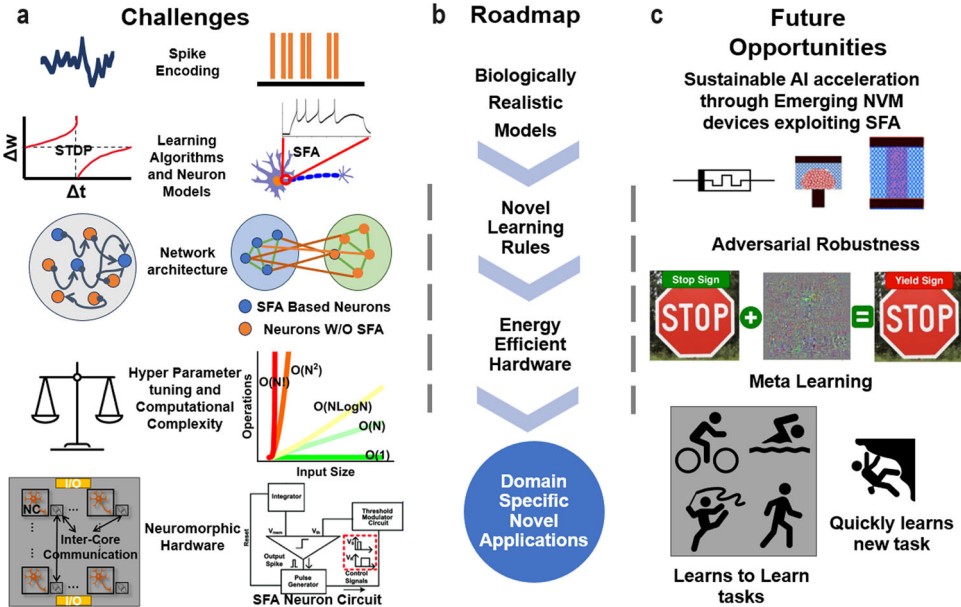

**Fig. 4 An Integrated Overview of SFA in Neuromorphic Computing.** The diagram unveils **a** the multifaceted open challenges yet to be thoroughly addressed for fully leveraging the merits of SFA, **b** outlines a methodological roadmap for the progressive development of SFA, and **c** pinpoints the promising avenues and untapped potential for future research and innovative applications.

involving higher-order synapse models like SRM, adds to the mathematical complexity. Innovative algorithmic refinement and numerical approximations tailored to SFA's unique characteristics are proposed as potential solutions, though further research is needed to confirm their effectiveness. Developing methods that capture essential dynamics without unnecessary computational overhead, specifically aligned with the nonlinear and stochastic elements of SFA, maybe a productive direction for reducing arithmetic demands. These suggestions represent possible paths for enhancing the adaptability and efficiency of SFA models but require rigorous testing and validation to determine their actual impact and viability.

*Integration and hardware compatibility.* Implementing SFA in contemporary digital and hybrid systems poses a nuanced challenge. In prevailing COTS neuromorphic computing platforms, LIF neurons are prevalent, often symbolizing a less biologically plausible approach. Though SFA can be attained using multi-compartment LIF neurons[31,52], this methodology might hinder efficiency in certain contexts. Striking an optimal balance between speed, footprint, and neuronal dynamics is an area demanding intensive exploration. The inherent challenges with analog circuits and scalability, particularly in analog circuit-based neurons, present substantial hurdles. Techniques focused on minimizing resource consumption through pre-synaptic spike-driven architecture may warrant comprehensive investigation to align with the progressing requirements of neuromorphic computing. This can significantly decrease the memory access. It's important to note that many state-of-the-art implementations utilize synchronous software models. However, asynchronous processing can potentially lead to further energy savings and computational advantages in SNNs. Future work may explore the integration of asynchronous mechanisms within these models to better align with biological neural systems

The advent of emerging technologies and the research in volatile resistive memory devices offer a promising frontier for the area-efficient development of adaptive neurons on analog hardware[86,87]. This approach can obviate the need for large capacitors, allowing the adjustment of time constants based on programming current, thus playing a crucial role in tuning the adaptation time constant for RSNNs.

The coming era may well witness robust advancements supporting SFA-based neurons, fostering a vibrant nexus between biological realism, technology, and emergent computational paradigms. However, the pathway is fraught with complexities related to scalability and the inherent challenges with analog circuits. The integration of these technologies may signify an essential step in enhancing the biological veracity and computational capacity of neuromorphic systems.

### Future opportunities
*Sustainable AI acceleration through emerging NVM devices exploiting SFA.* A compelling avenue for future exploration lies in the convergence of SFA and emerging NVM technologies to propel the development of next-generation, sustainable AI hardware. Notably, recent research[19] has demonstrated that the nonlinear conductance changes intrinsic to NVM devices can be harnessed as a mechanism for threshold adaptation in SFA neurons. By capitalizing on this synergy, AI hardware can tap into the inherent adaptiveness of SFA to dynamically modulate neural responses. The programmable threshold behavior, facilitated by NVM's non-linear conductance change, aligns seamlessly with SFA's temporal sensitivity. This tandem approach not only enhances energy efficiency by eliminating the need for static threshold levels but also fosters inherent fault tolerance, mitigating variations in NVM devices. Furthermore, the incorporation of SFA-based NVM hybrid systems holds promise for constructing highly efficient memory and energy architectures. The adaptability of SFA can enable selective information filtering, thereby minimizing memory access and bolstering resource efficiency. NVM's natural properties, integrated with SFA, pave the way for optimized AI accelerators that balance performance, energy consumption, and memory utilization.

*Real-time adaptation in dynamic environments.* SFA's intrinsic ability to prevent neural saturation ensures that SNNs remain sensitive to fluctuating environments. For critical real-time

applications such as autonomous vehicles and robotic systems, the incorporation of SFA may offer novel strategies for achieving both instantaneous adaptability and long-term stability. This facet of SFA could lead to groundbreaking advancements in real-time decision-making algorithms and adaptive control systems.

*Continuous learning and temporal feature extraction.* The inclusion of adaptation in neuron models through SFA introduces longer time constants that may significantly aid in learning temporal features of the input. By exploiting the extra available time scales, the network can enhance online learning convergence time and provide a more nuanced understanding of temporal dynamics. This could foster advancements in speech recognition, time-series prediction, and online learning systems, where temporal relationships are essential.

*Enhanced robustness against adversarial attacks.* Recent studies have underscored the inherent resilience of SNNs to specific adversarial perturbations[88–91]. The selective responsiveness of SFA to changes in input layers, acting as a form of firewall, could further amplify this robustness. This opens avenues to develop advanced defenses against adversarial attacks and contributes to the fortification of network security. Implementing SFA in robust models may lead to novel mechanisms to mitigate threats in cybersecurity.

*Regularization and meta-learning.* SFA's ability to adapt to input frequency could serve as a form of regularization, potentially preventing overfitting in deep learning scenarios. In the context of meta-learning, where catastrophic forgetting is a significant concern[92,93], SFA's adaptive thresholds may enable the network to discern underlying patterns across different tasks. This adaptation process may play a vital role in solving complex meta-learning challenges, including multi-modal learning and cross-domain adaptation, thereby aligning with advanced research directions.

The exploration of synergies between SNNs, SFA neurons, and NVM technologies presents intriguing possibilities. While still at an experimental stage, the future opportunities discussed offer a glimpse of potential pathways that could contribute to more efficient, robust, and adaptive AI systems. These innovations might shape the next phase of computational intelligence, yet their realization will depend on sustained research, collaboration, and a keen understanding of the complex interplay between these cutting-edge technologies.

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

## Acknowledgements

S.S.B.'s work was partially supported by the USA National Science Foundation award #2318152. E.A. was supported by the European Union's Horizon 2020 research and innovation program under the Marie Skłodowska-Curie grant agreement No. 101031746. M.P. was supported by SNSF Starting Grant Project UNITE (TMSGI2-211461).

## Author contributions

C.G., S.S.B., S.D., M.S conceptualized the review. C.G., S.S.B. prepared the original draft, and S.S.B. lead on the manuscript revisions. E.A. enriched the content with a neuroscience perspective, authored specific segments, and participated in reviews. M.P., S.D., and M.S. provided key insights, contributed to particular subsections, and aided in manuscript refinement.

## Competing interests

The authors declare no competing interests.
