## [Peer Review File · Communications Engineering]

Reviewers' comments:

Reviewer #1 (Remarks to the Author):

The paper does not introduce the motivation for adaptive spiking neurons. It only mentions brain inspiration, but the description lacks details and needs to be more satisfactory (i.e., it needs to clarify what brain properties of adaptable neurons have a computational advantage). Moreover, the link between power efficiency and spike frequency adaptation must be clarified.

Several adaptive neuron models exist, and the choice of the selected and described one is not discussed. Section III of the paper presents several neural models, but no pros, contra, or tradeoffs are described; it is a list of different models, and therefore, the main objective of this section needs to be clarified.

Section IV lists recent results on state-of-the-art studies with ALIF in SNN. Many of these studies do not employ (i) asynchronous data processing, (ii) sparse activations, and (iii) do not provide energy efficiency compared to traditional DNN models. These objectives are the paper's main objective in the introduction and should come back in this section. However, only the accuracy and number of flops required are listed in this section. Moreover, it is not clear how to relate the number of flops with energy consumption. How does the number of flops related to power? Furthermore, what about asynchronicity (many of the implementations reported are synchronous software models)?

It seems that flops are helpful for software-based implementation and are not descriptive of neuromorphic implementations in analog and emerging substrates. This point needs clarification (relationship between flops and energy). As pointed out by the authors, several implementations have been presented in the literature, but the paper does not provide a comprehensive comparison.

Section V is very short, and also, it is a list of a few selected papers that need a clear justification for why these papers have been selected. Moreover, the tradeoffs could be more detailed.

Section VI, while I agree with many of the challenges, it seems there is little organization for this section. For example 6G future communications systems and wireless technologies are mentioned without a clear link with adaptive spiking neural networks.

Section VI.c is a list of 4 points without discussion.

In addition: the equations require the definition of all of the variables (e.g., eq 1, what is R ? and why does it become R_m in equation 2? what is v_{rest} (it is only introduced in eq 7 but it appears in eq 2)? etc.!

In conclusion, the paper does not provide a comprehensive review of adaptive neuron models nor the fundamental limitations and or tradeoffs of adaptive spiking neural networks. The problem needs to be better articulated, and the long-term vision needs to be more precise; only broad claims and a list of results are reported. The conclusion and roadmap section do not envision any breakthroughs but list some possible explorations without explaining the significance and without prioritizing them in order of importance or impact they could have.

In summary, I suggest rejection of this work. The paper would benefit from a clear structure and content organization.

Reviewer #2 (Remarks to the Author):

The authors propose a survey of computational models of spikes frequency adaptation along with their hardware implementations and potential applications.

The survey is relevant and might contribute to the ongoing discussions in the neuromorphic community. The survey is well structured, going from biological description to computational modeling, hardware realization, and applications. I particularly appreciate their take on simulators' ability to support frequency adaptation.

Few notes:

1. The manuscript title is a bit misleading. It is not clear what makes these neurons "advance". I would eliminate it. Also, the term "adaptive spiking neuron" is too general, as the authors are focusing solely on frequency adaptation. I would change the title to reflect it.
2. The authors might want to extend their discussion on hardware implementation in CMOS analog circuits, which is currently quite limited in scope. I refer the authors to DOI: 10.1109/BioCAS49922.2021.9644944.

Reviewer #3 (Remarks to the Author):

This paper presents a review of some of the currently available adaptive neuron models used in spiking neural networks. It adds context to these models by providing examples of where the models have been implemented in both hardware and simulation. Finally, the authors present several ideas for how these models can be developed and applied in the future.

As the paper is a review, novelty is not expected. However, the paper would benefit from a clear vision so that the reader understands why a review of adaptive spiking models is relevant for scientists in the field and wider community. In the introduction, the review claims that adaptive spiking neurons will increase both computational and power efficiency. If this is the main claim of the review, it would be helpful to reinforce this point throughout the paper and elaborate on it again in the "road map" and "conclusion". Efficiency is mentioned in the "new research opportunities" so the authors can consider elaborating on this point within that section. The authors should also cite the recent Furber paper (Furber, S., & Davidson, S. (2021). Comparison of Artificial and Spiking Neural Networks on Digital Hardware. *Frontiers in Neuroscience*, 15, 651141. [651141]. <https://doi.org/10.3389/fnins.2021.651141>) which compares efficiency and argues that spiking neural networks are not currently more efficient than DNNs. Even though they indicate that SNNs are not necessarily efficient, the paper can be used to support the claim that spike adaptation specifically is an efficiency mechanism to be exploited.

Section III. "Description of Adaptive Neuron Models" would benefit from a figure or table that places

these neuron models in relation to each other to understand how they differ and what the benefits of a particular model are and/or for which application it is best suited. The tables in subsequent sections are helpful and provide the reader with a quick overview / main takeaways of the section; this type of overview would help Section III.

More generally, the flow of the writing can be difficult to follow. Some paragraphs consist of short sentences that can come across as unrelated. The authors should closely review the paper to improve both the flow and the grammar.

Example of flow: “The learning algorithm used in [10, 12] is backpropagation through time (BPTT). A learning algorithm, called e-prop for recurrent spiking neural networks (RSNN), which is alternative to BPTT is proposed in [13].”

This could be changed to:

[10, 12] use backpropagation through time (BPTT) as a learning algorithm. This algorithm is characterized by... A further development was made by [13] where they propose e-prop as an alternative to BPTT. This differs from BPTT as it uses...

Example of grammar:

“In [35], author proposed an adaptive threshold module (ATM) for a SNN based architecture. ATM algorithm controls internal threshold potential. This ATM is used to control output firing rate, which helps to to extract the information encoded in input stimulus.”

Corrected:

In [35], the author proposed an adaptive threshold module (ATM) for an SNN based architecture. An ATM algorithm controls internal threshold potential. This ATM is used to control output firing rate, which helps to extract the information encoded in the input stimulus.

Minor comments: The authors should consider citing reference [55] when introducing the adaptive LIF in the introduction.

The abstract states that neurons only communicate through spikes. This should be modified to state that one of the methods of communication is spikes. Neurons can also communicate on a sub-threshold level through, for example, electrical synapses.

Overall, the paper provides a good overview of adaptive spiking neuron models for scientists that may be entering the field. However, the paper requires significant improvement in order for the reader to understand the relevance of this work in the current scientific paradigm. This can be accomplished by reinforcing the main claim of the paper - that adaptive spiking models improve efficiency - throughout all of the sections. By relating each of the sections back to the overarching theme, it will provide a storyline for the reader to follow.

Reviewer #4 (Remarks to the Author):

Major revision

Comments for Survey paper on neuron models for SNNs

This paper surveys different adaptive neuron models for their use in Spiking Neural Networks. It starts justifying the benefits of adaptive models, and it then reviews the models to be reviewed. From this point on, the paper examines several case studies and some hardware implementations. In the end, a road map is suggested.

There is a number of major/minor comments from this reviewer's point of view, that support my suggestion of not accepting the paper in its current status:

- Is not appropriate to refer to some other SOA contributions related to the field, like those from Simon Thorpe or W. Mass?
- In the section where you review the adaptive models, you present the formulation for almost all of them with $u(t)$ and $v_{th}(t)$, but not in all of them. I recommend presenting all in the same way and keep highlighting the differences in the formulation between the models.
- Just before eq 17 you state: "The set F is collection of all spike times...". It seems you missed the article for collection (a, the)
- Comment: in the SRM equation for $u(t)$ there is a power factor of g in t_j , does it means that the more spikes, the higher the threshold?
- The nomenclature convention used in the first paragraph of model G: generalized LIF, is different from the one used later: there is no reference to GLIF-I in that first paragraph, so numbering is different afterward.
- Typo in section IV, paragraph "Multiple spatio-temporal...": STORE and RECALL task. Shouldn't it be "tasks"? Later in this paragraph, you state: 10 LIF and 10 DEXAT neurons are used for the task. What task are you referring to?
- A summary table before section VI offering the model, its reference, the simulator used and the tool of framework used would be very interesting.
- Fig 4: Why is the architecture of [30] interesting and not the others? In a survey paper, shouldn't it be interesting to highlight the differences between mentioned network architectures for different reviewed papers?
- First paragraph of page 11: hyperparameter tuning: Have you detected a relation between application examples of the covered literature and the architecture and parameters of the SNN used?
- At the end of section VI.B you offer a list of possible applications where these models with SNN can be beneficial. This part should be expanded a bit with deeper definitions, clarifications, justifications, and discussions.
- The list of new research opportunities requires major review: a list is not adequate. Expand and discuss. Give your opinion based on collected information from literature and what you see could be the future.
- The conclusion section is vague. Expand and provide concrete conclusions on your survey.
- Figure 5-a: why organic roadmap? What is the reason for using the word organic here?
- Title should be less generalistic since your review is not so broad in general terms (implementations for example)

Reviewer #1 (Remarks to the Author):

The paper does not introduce the motivation for adaptive spiking neurons. It only mentions brain inspiration, but the description lacks details and needs to be more satisfactory (i.e., it needs to clarify what brain properties of adaptable neurons have a computational advantage). Moreover, the link between power efficiency and spike frequency adaptation must be clarified.-

Thank you for your insightful comments and observations. In the current version of manuscript we have emphasized the role of SFA in reducing memory access, consequently improving computational efficiency. We also related this reduction to a decrease in both area and energy footprint, with references to studies that demonstrate the benefits of SFA.

"The memory bottleneck in neural computation must be carefully considered, as memory access often consumes more time than computation itself, according to [15]. Within this context, the role of SFA becomes pertinent. Due to SFA, there is a decrease in spike frequency, leading to a corresponding reduction in the number of synaptic memory accesses, which are contingent on a prespike from the preceding layer. When compared to Leaky Integrate-and-Fire (LIF) models, this reduction in spikes has the potential to decrease computational inefficiency. Further studies, as referenced in [12, 16, 17], demonstrate that the benefits of SFA allow for a reduction in the number of neurons required to achieve similar accuracy in various spatio-temporal tasks. This reduction contributes to a decrease in both the area and energy footprint for the corresponding application."

Several adaptive neuron models exist, and the choice of the selected and described one is not discussed. Section III of the paper presents several neural models, but no pros, contra, or tradeoffs are described; it is a list of different models, and therefore, the main objective of this section needs to be clarified.

Thank you for your insightful comments and observations.

In this review paper, we have considered adaptive models based on the premise of the LIF framework.

Leaky integrate-and-fire (LIF) models are popular in spiking neural networks (SNNs) due to their simplicity, computational efficiency, and ability to capture some essential aspects of temporal character. Due to their simplicity, LIF models are amenable to theoretical analysis, making them useful for studying fundamental properties of spiking neural networks, such as stability, dynamics, and network analysis. It is important to note that leaky integrate-and-fire models have their advantages but are still simplified abstractions of real neurons. An essential feature of a neuron missing in LIF is spike frequency adaptation.

ALIF models encompass all benefits of a LIF model and use a dynamic threshold that changes based on the neuron's recent activity. This mechanism can lead to more sophisticated information processing, as the neuron's sensitivity to input can be modulated by its recent firing history. With a more complex adaptation mechanism, the model attains higher efficiency with

less iteration [11, 16, 24]. The GLIF models are listed in ascending order of complexity. As model complexity escalates, it requires more arithmetic operations per iteration, as listed in Table I. ALIF can replicate the phenomenon of spike-frequency adaptation, where neurons become less responsive to repeated input spikes over time. This feature allows SNNs to capture more nuanced response patterns and better represent certain types of neural processing, increasing the computational efficiency as proven in the paper [11, 16, 24]. ALIF models can be easily combined with synaptic plasticity rules to study learning and memory processes in spiking neural networks. The adaptive behaviour of these models allows for a more realistic exploration of synaptic strength changes and their impact on network function. ALIF models can also be implemented on neuromorphic hardware platforms, taking advantage of their more biologically plausible nature.

The above section has now been added in the section III of the modified manuscript. Table I also has been modified with corresponding membrane potential and threshold adaptation equations.

Section IV lists recent results on state-of-the-art studies with ALIF in SNN. Many of these studies do not employ (i) asynchronous data processing, (ii) sparse activations, and (iii) do not provide energy efficiency compared to traditional DNN models. These objectives are the paper's main objective in the introduction and should come back in this section. However, only the accuracy and number of flops required are listed in this section. Moreover, it is not clear how to relate the number of flops with energy consumption. How does the number of flops related to power? Furthermore, what about asynchronicity (many of the implementations reported are synchronous software models)?

Thank you for your insightful comments and observations regarding Section IV, specifically on the aspects of asynchronous data processing, sparse activations, energy efficiency, and the relationship between the number of flops and power consumption. In response to these points, we have made the following revisions to provide a more comprehensive view:

SFA in Computation: We highlighted that Spike Frequency Adaptation (SFA) is vital for computation through spiking neurons, especially in overcoming challenges in liquid state machines or recurrent neural networks. This function, enabled through an adaptive threshold, allows SFA neurons in Spiking Neural Networks (SNNs) to reach accuracy levels comparable to ANN-LSTMs.

"Authors showed that SFA is crucial as well for computation through spiking neurons. This function is particularly instrumental in overcoming the vanishing gradient problem in liquid state machines or recurrent neural networks through the employment of an adaptive threshold, which serves as the source of SFA, within the gradient calculation process. Furthermore, the studies provided evidence that SNNs equipped with SFA neurons are capable of achieving accuracy levels comparable to those found in ANN-LSTMs."

Memory Bottleneck and Energy Efficiency: We have emphasized the role of SFA in reducing memory access, consequently improving computational efficiency. We also related this reduction to a decrease in both area and energy footprint, with references to studies that demonstrate the benefits of SFA.

"The memory bottleneck in neural computation must be carefully considered, as memory access often consumes more time than computation itself, according to [15]. Within this context, the role of SFA becomes pertinent. Due to SFA, there is a decrease in spike frequency, leading to a corresponding reduction in the number of synaptic memory accesses, which are contingent on a prespike from the preceding layer. When compared to Leaky Integrate-and-Fire (LIF) models, this reduction in spikes has the potential to decrease computational inefficiency. Further studies, as referenced in [12, 16, 17], demonstrate that the benefits of SFA allow for a reduction in the number of neurons required to achieve similar accuracy in various spatio-temporal tasks. This reduction contributes to a decrease in both the area and energy footprint for the corresponding application."

Asynchronicity: In the context of our study, we acknowledge that many of the implementations reported are synchronous models. Asynchronous data processing represents an important dimension in spiking neural network (SNN) efficiency, especially in neuromorphic computing. While our paper primarily focuses on other aspects, we recognize the significance of asynchronicity, and we have added the following text to Section VI:

"It's important to note that many state-of-the-art implementations utilize synchronous software models. However, asynchronous processing can potentially lead to further energy savings and computational advantages in SNNs. Future work may explore the integration of asynchronous mechanisms within these models to better align with biological neural systems."

In order to avoid confusion, FLOP in the modified manuscript is updated to the number of arithmetic operations per iteration.

Relation to Power: The relationship between the number of arithmetic operations per iteration and energy consumption is indeed complex and multifaceted. The number of arithmetic operations per iteration can be indicative of the computational workload, but it doesn't directly translate to power consumption. Factors such as hardware architecture, operating frequency, data movement, and memory access play key roles in determining energy consumption. We have added a paragraph to clarify this relationship in Section III of the modified manuscript:

"While the number of arithmetic operations per iteration required is often used as a proxy for computational complexity, it's essential to recognize that it doesn't linearly correlate with power consumption. Energy efficiency depends on various other factors, including hardware design, memory access patterns, and algorithmic optimizations."

We believe that these additions respond to your comments and provide clarity on the mentioned topics. We thank you for bringing these important aspects to our attention and guiding us in enhancing the depth of our paper.

It seems that flops are helpful for software-based implementation and are not descriptive of neuromorphic implementations in analog and emerging substrates. This point needs clarification (relationship between flops and energy). As pointed out by the authors, several implementations have been presented in the literature, but the paper does not provide a comprehensive comparison.

Thank you for pointing out the distinction between operations in software-based implementations and the unique considerations in analog circuits and emerging substrates. We agree that FLOPS is not an appropriate metric in these contexts.

In our revised manuscript, we have updated our terminology from "flops" to "operations," recognizing that multiplication and addition can be performed more easily in analog circuits. Moreover, we've expanded on how the natural non-idealities in emerging substrates can be harnessed to perform complex calculations with greater efficiency.

Section V is very short, and also, it is a list of a few selected papers that need a clear justification for why these papers have been selected. Moreover, the tradeoffs could be more detailed.

Thank you for your comment regarding Section V. In response to your concerns, we have expanded this section to include a broader selection of papers, adding 13 new references that encompass various approaches to Spike Frequency Adaptation (SFA) integration within hardware.

We've also enriched the discussion on the trade-offs associated with different technological solutions, with a focus on design complexity, efficiency, and potential optimization in resource utilization. These enhancements aim to provide a more comprehensive perspective on the multidimensional potential of SFA within neuromorphic hardware.

The updated section has been included, and we believe it addresses your concerns by offering a more detailed and justified examination of recent developments in the field. Thank you once again for your constructive feedback.

Section VI, while I agree with many of the challenges, it seems there is little organization for this section. For example 6G future communications systems and wireless technologies are mentioned without a clear link with adaptive spiking neural networks.

Section VI.c is a list of 4 points without discussion.

Thank you for your insightful comment regarding the organization of Section VI and the mention of "6G communication systems and wireless technologies."

In our initial draft, we included 6G as an attempt to explore possible intersections between the adaptive nature of Spike Frequency Adaptation (SFA) and emerging communication technologies. However, we acknowledge that this connection was not well-supported in our text.

Based on your feedback, we have removed the reference to 6G from the revised manuscript. Additionally, we have restructured Section VI into two more concise and focused sub-sections:

A. Challenges and Roadmap - Covering Encoding Techniques in SFA, Learning Algorithms and Adaptive Neurons, Network Architecture and Connectivity, Hyperparameter Tuning and Mathematical Complexity of SFA Models, and Integration and Hardware Compatibility.

B. Future Opportunities - Including Sustainable AI acceleration through Emerging NVM devices exploiting SFA, Real-Time Adaptation in Dynamic Environments, Continuous Learning and Temporal Feature Extraction, Enhanced Robustness Against Adversarial Attacks, and Regularization and Meta-Learning.

We believe these revisions provide greater clarity and focus in presenting the current challenges and future opportunities of SFA. Thank you once again for your valuable input.

In addition: the equations require the definition of all of the variables (e.g., eq 1, what is R ? and why does it become R_m in equation 2? what is v_{rest} (it is only introduced in eq 7 but it appears in eq 2)? Etc.!

Thank you for pointing out the errors.

The notations have been corrected for uniformity, and corresponding symbol definitions have now been included in the modified manuscript.

In conclusion, the paper does not provide a comprehensive review of adaptive neuron models nor the fundamental limitations and or tradeoffs of adaptive spiking neural networks. The problem needs to be better articulated, and the long-term vision needs to be more precise; only broad claims and a list of results are reported. The conclusion and roadmap section do not envision any breakthroughs but list some possible explorations without explaining the significance and without prioritizing them in order of importance or impact they could have.

In summary, I suggest rejection of this work. The paper would benefit from a clear structure and content organization.

Reviewer #2 (Remarks to the Author):

The authors propose a survey of computational models of spikes frequency adaptation along with their hardware implementations and potential applications.

The survey is relevant and might contribute to the ongoing discussions in the neuromorphic community.

The survey is well structured, going from biological description to computational modeling, hardware realization, and applications. I particularly appreciate their take on simulators' ability to support frequency adaptation.

Few notes:

1. The manuscript title is a bit misleading. It is not clear what makes these neurons "advance". I would eliminate it. Also, the term "adaptive spiking neuron" is too general, as the authors are focusing solely on frequency adaptation. I would change the title to reflect it

Thank you for your insightful comment. We agree with your observation and have revised the title to "Spike Frequency Adaptation: Bridging Neural Models and Neuromorphic Applications," to better reflect the focus of the paper.

2. The authors might want to extend their discussion on hardware implementation in CMOS analog circuits, which is currently quite limited in scope. I refer the authors to DOI: 10.1109/BioCAS49922.2021.9644944.

Thank you for the insightful comment regarding the extension of our discussion on hardware implementation. In response, we have thoroughly expanded the section on the integration of Spike Frequency Adaptation (SFA) models within hardware. This includes an in-depth examination of existing commercial platforms, recent developments in neuromorphic computing, and a broader analysis of hardware implementations encompassing CMOS and emerging resistive memory technologies. 9 new references have been added to ensure a comprehensive overview. We believe that these modifications add substantial depth to our paper, aligning with your valuable recommendation.

"The integration of Spike Frequency Adaptation (SFA) models within hardware has progressively manifested as a seminal approach to augmenting the efficiency of AI hardware, with promising applications in neuromorphic computing. Existing commercial off-the-shelf (COTS) platforms, deploying Leaky Integrate-and-Fire (LIF) neuron blocks as fundamental units, have seen extensive research for implementing SFA neurons in multicompartment neuron configurations [16, 40–42].

Recent developments in the field, such as the work by [16], have proven the feasibility of achieving resource utilization with reduced neuron count. Further, Intel's Loihi2 architecture has ventured into adding ALIF models, heralding a promising avenue in COTS hardware. Parallel to these advancements, hardware implementations of neuron models such as Integrate and Fire (IF) and LIF [43–45] have been widely reported, encompassing complementary metal oxide semiconductor (CMOS) and emerging resistive memory technologies like RRAM, PCM, and CBRAM [46–55]. Recently, superconducting device, 2D material based device neuron circuits had shown SFA [56, 57].

Digital implementation of modified AdEx neuron models on FPGA further amplifies the possibilities [58–60]. Innovations such as [61] demonstrate improvements in speed and footprint

without compromising neuronal dynamics. The utilization of quantized versions of DEXAT neuron models [12] represents another noteworthy advancement. Notably, the integration of SFA within FPGA has led to the development of a presynaptic spike-driven architecture, which significantly reduces resource utilization and buffer size for caching events, while maintaining accurate task-solving performance [62].

The confluence of these developments underlines the multidimensional potential of SFA within neuromorphic hardware. The exploration of digital circuits, analog designs, and emerging NVM devices presents a diverse spectrum of opportunities and challenges. The emergence of space-efficient and low-power circuits constructed with advanced 3D integration technologies indicates the path forward.

In conclusion, the adoption and adaptation of SFA within neuromorphic hardware demonstrate a forward-thinking approach in both design complexity and efficiency. This integration harbors significant potential not only in optimizing resource utilization.”

Reviewer #3 (Remarks to the Author):

This paper presents a review of some of the currently available adaptive neuron models used in spiking neural networks. It adds context to these models by providing examples of where the models have been implemented in both hardware and simulation. Finally, the authors present several ideas for how these models can be developed and applied in the future.

As the paper is a review, novelty is not expected. However, the paper would benefit from a clear vision so that the reader understands why a review of adaptive spiking models is relevant for scientists in the field and wider community. In the introduction, the review claims that adaptive spiking neurons will increase both computational and power efficiency. If this is the main claim of the review, it would be helpful to reinforce this point throughout the paper and elaborate on it again in the “road map” and “conclusion”. Efficiency is mentioned in the “new research opportunities” so the authors can consider elaborating on this point within that section.

Thank you for emphasizing the need for clarity on the relevance of spike frequency adaptation (SFA) and its connection to computational and power efficiency. In response to your suggestion, we have undertaken significant revisions in Section VI, dividing it into two focused sub-sections:

A. Challenges and Roadmap - This covers critical topics such as Encoding Techniques in SFA, Learning Algorithms and Adaptive Neurons, Network Architecture and Connectivity, Hyperparameter Tuning and Mathematical Complexity of SFA Models, and Integration and Hardware Compatibility.

B. Future Opportunities - Here, we explore areas like Sustainable AI acceleration through Emerging NVM devices exploiting SFA, Real-Time Adaptation in Dynamic Environments,

Continuous Learning and Temporal Feature Extraction, Enhanced Robustness Against Adversarial Attacks, and Regularization and Meta-Learning.

Additionally, under the section "Why adaptive neuron models?", we have attempted to further highlight how SFA might help address the memory bottleneck and the vanishing gradient problem in RSNNs. We hope that these revisions contribute to reinforcing the main claim of computational and power efficiency throughout the paper and align more closely with the vision you suggested for this review.

The authors should also cite the recent Furber paper (Furber, S., & Davidson, S. (2021). Comparison of Artificial and Spiking Neural Networks on Digital Hardware. *Frontiers in Neuroscience*, 15, 651141. [651141]. <https://doi.org/10.3389/fnins.2021.651141>) which compares efficiency and argues that spiking neural networks are not currently more efficient than DNNs. Even though they indicate that SNNs are not necessarily efficient, the paper can be used to support the claim that spike adaptation specifically is an efficiency mechanism to be exploited.

Thank you for pointing out the recent Furber paper. We agree that it offers a pertinent perspective on the efficiency of Spiking Neural Networks. In light of your suggestion, we have included this reference in our manuscript, specifically in the sections discussing spike adaptation's potential as an efficiency mechanism.

Section III. "Description of Adaptive Neuron Models" would benefit from a figure or table that places these neuron models in relation to each other to understand how they differ and what the benefits of a particular model are and/or for which application it is best suited. The tables in subsequent sections are helpful and provide the reader with a quick overview / main takeaways of the section; this type of overview would help Section III.

Thank you for your insightful comments and observations.

In this review paper, we have considered adaptive models based on the premise of the LIF framework.

Leaky integrate-and-fire (LIF) models are popular in spiking neural networks (SNNs) due to their simplicity, computational efficiency, and ability to capture some essential aspects of temporal character. Due to their simplicity, LIF models are amenable to theoretical analysis, making them useful for studying fundamental properties of spiking neural networks, such as stability, dynamics, and network analysis. It is important to note that leaky integrate-and-fire models have their advantages but are still simplified abstractions of real neurons. An essential feature of a neuron missing in LIF is spike frequency adaptation.

ALIF models encompass all benefits of a LIF model and use a dynamic threshold that changes based on the neuron's recent activity. This mechanism can lead to more sophisticated

information processing, as the neuron's sensitivity to input can be modulated by its recent firing history. With a more complex adaptation mechanism, the model attains higher efficiency with less iteration [11, 16, 24]. The GLIF models are listed in ascending order of complexity. As model complexity escalates, it requires more arithmetic operations per iteration, as listed in Table I. ALIF can replicate the phenomenon of spike-frequency adaptation, where neurons become less responsive to repeated input spikes over time. This feature allows SNNs to capture more nuanced response patterns and better represent certain types of neural processing, increasing the computational efficiency as proven in the paper [11, 16, 24]. ALIF models can be easily combined with synaptic plasticity rules to study learning and memory processes in spiking neural networks. The adaptive behaviour of these models allows for a more realistic exploration of synaptic strength changes and their impact on network function. ALIF models can also be implemented on neuromorphic hardware platforms, taking advantage of their more biologically plausible nature.

The above section has now been added to Section III of the modified manuscript. As per suggestion, Table I also has been modified with the corresponding membrane potential and threshold adaptation equations.

More generally, the flow of the writing can be difficult to follow. Some paragraphs consist of short sentences that can come across as unrelated. The authors should closely review the paper to improve both the flow and the grammar.

Example of flow: “The learning algorithm used in [10, 12] is backpropagation through time (BPTT). A learning algorithm, called e-prop for recurrent spiking neural networks (RSNN), which is alternative to BPTT is proposed in [13].”

This could be changed to:

[10, 12] use backpropagation through time (BPTT) as a learning algorithm. This algorithm is characterized by... A further development was made by [13] where they propose e-prop as an alternative to BPTT. This differs from BPTT as it uses...

Example of grammar:

“In [35], author proposed an adaptive threshold module (ATM) for a SNN based architecture. ATM algorithm controls internal threshold potential. This ATM is used to control output firing rate, which helps to to extract the information encoded in input stimulus.”

Corrected:

In [35], the author proposed an adaptive threshold module (ATM) for an SNN based architecture. An ATM algorithm controls internal threshold potential. This ATM is used to control output firing rate, which helps to extract the information encoded in the input stimulus.

Thank you for the suggestion.

The entire manuscript has been thoroughly revised to eliminate grammatical mistakes and improve readability.

Minor comments: The authors should consider citing reference [55] when introducing the adaptive LIF in the introduction.

The abstract states that neurons only communicate through spikes. This should be modified to state that one of the methods of communication is spikes. Neurons can also communicate on a sub-threshold level through, for example, electrical synapses.

Thank you for your insightful comment and observation.
However, we consider the neurons connected through chemical synapses mainly. This has now been clearly mentioned in the abstract.

Overall, the paper provides a good overview of adaptive spiking neuron models for scientists that may be entering the field. However, the paper requires significant improvement in order for the reader to understand the relevance of this work in the current scientific paradigm. This can be accomplished by reinforcing the main claim of the paper - that adaptive spiking models improve efficiency - throughout all of the sections. By relating each of the sections back to the overarching theme, it will provide a storyline for the reader to follow.

Reviewer #4 (Remarks to the Author):

Major revision

Comments for Survey paper on neuron models for SNNs

This paper surveys different adaptive neuron models for their use in Spiking Neural Networks. It starts justifying the benefits of adaptive models, and it then reviews the models to be reviewed. From this point on, the paper examines several case studies and some hardware implementations. In the end, a road map is suggested.

There is a number of major/minor comments from this reviewer's point of view, that support my suggestion of not accepting the paper in its current status:

- Is not appropriate to refer to some other SOA contributions related to the field, like those from Simon Thorpe or W. Mass?

Thank you for your comment. Some of the recent works of W. Mass have already been referred to in the manuscript. The works of W. Mass that are referred to in the manuscript are listed below:

[10] D. Salaj, A. Subramoney, C. Krausnikovic, G. Bellec, R. Legenstein, and W. Maass, "Spike frequency adaptation supports network computations on temporally dispersed information," *Elife*, vol. 10, p. e65459, 2021

- [15] G. Bellec, D. Salaj, A. Subramoney, R. Legenstein, and W. Maass, "Long short-term memory and learning-to-learn in networks of spiking neurons," arXiv preprint arXiv:1803.09574 , 2018.
- [16] G. Bellec, F. Scherr, A. Subramoney, E. Hajek, D. Salaj, R. Legenstein, and W. Maass, "A solution to the learning dilemma for recurrent networks of spiking neurons," Nature communications, vol. 11, no. 1, p. 3625, 2020.
- [22] Subramoney, G. Bellec, F. Scherr, R. Legenstein, and W. Maass, "Revisiting the role of synaptic plasticity and network dynamics for fast learning in spiking neural networks," bioRxiv , 2021
- [24] G. Bellec, F. Scherr, A. Subramoney, E. Hajek, D. Salaj, R. Legenstein, and W. Maass, "A solution to the learning dilemma for recurrent networks of spiking neurons," Nature communications, vol. 11, no. 1, pp. 1–15, 2020
- [36] A. Rao, P. Plank, A. Wild, and W. Maass, "A long short-term memory for ai applications in spike-based neuromorphic hardware," Nature Machine Intelligence, vol. 4, no. 5, pp. 467–479, 2022

- In the section where you review the adaptive models, you present the formulation for almost all of them with $u(t)$ and $v^h(t)$, but not in all of them. I recommend presenting all in the same way and keep highlighting the differences in the formulation between the models.

- Just before eq 17 you state: "The set F is collection of all spike times...". It seems you missed the article for collection (a, the)

Thank you for pointing out the error. It has been corrected in the modified manuscript.

- Comment: in the SRM equation for $u(t)$ there is a power factor of g in t_j , does it means that the more spikes, the higher the threshold? Dr. Chittotosh

Thank you for your comment. The term " g " is not an exponent in equation 20. Term $t_j^{(g)}$

denotes spike time of g^{th} spike from j^{th} pre-synaptic neuron.

- The nomenclature convention used in the first paragraph of model G: generalized LIF, is different from the one used later: there is no reference to GLIF-I in that first paragraph, so numbering is different afterward.

Thank You for pointing out the error. GLIF abbreviation is now added at the beginning of the sub-section. Five different variants of GLIFs are mentioned in the modified manuscript as "Five GLIF models are found in the literature, namely GLIF-I to GLIF-V."

- Typo in section IV, paragraph "Multiple spatio-temporal...": STORE and RECALL task. Shouldn't it be "tasks"? Later in this paragraph, you state: 10 LIF and 10 DEXAT neurons are used for the task. What task are you referring to?

Thank you for identifying the typo in section IV. You are correct, it should indeed read "STORE and RECALL tasks." We appreciate your keen observation.

Regarding your question about the task referred to in the statement "10 LIF and 10 DEXAT neurons are used for the task," we are referring to a specific experiment where these neuron models were utilized. In the revised manuscript, we have clarified this part by providing additional context and details about the particular task, ensuring that the description is more precise and informative.

The corrected section now reads: "Multiple spatio-temporal pattern recognition tasks, including STORE and RECALL tasks, were conducted in our experiment. For these specific tasks, 10 LIF and 10 DEXAT neurons were used."

We hope that these modifications resolve the confusion and provide a clear understanding of the concepts discussed in that section.

- A summary table before section VI offering the model, its reference, the simulator used and the tool of framework used would be very interesting.

We appreciate your suggestion to include a summary table before section VI detailing the model, its reference, the simulator used, and the tool or framework used. We recognize the value this would add to our paper.

Unfortunately, during our review of the referenced papers, we were unable to find specific information about the simulators used in each case. However, we have made a concerted effort to enhance our simulator section by providing information on various simulators that support adaptive neuron models for building spiking neural networks, such as PyNN, NEST, Neko, FABLE, and Norse.

We have included three new references to ensure that this section is both current and comprehensive, as shown in the following modification:

"Various simulators support adaptive neuron models for building spiking neural networks. The function of the AdEx neuron model is based on polarizing and hyperpolarizing currents supported by PyNN [65] and NEST [66]. Neko [67], FABLE [68], and Norse [69] are SNN simulation frameworks based on PyTorch that enable the ALIF neuron model for constructing Recurrent-SNN. Here, the ALIF neuron model is a state function in which the membrane voltage and neuron threshold are updated with every iteration. More hardware realistic neuromorphic circuit simulation is shown in [60]."

- Fig 4: Why is the architecture of [30] interesting and not the others? In a survey paper, shouldn't it be interesting to highlight the differences between mentioned network architectures for different reviewed papers?

Thank you for highlighting the importance of presenting a comprehensive comparison of different network architectures. We concur that a detailed examination of various architectures

would enhance our survey's value. Therefore, we have removed Figure 4, which was not contributing significantly to readers' understanding.

Instead, we have expanded the discussion section to elaborate on the architectures generally explored in the context of Spike Frequency Adaptation (SFA), namely Feed Forward and Recurrent Spiking Neural Networks (SNNs). We have also delved into the exciting possibilities and challenges associated with integrating diverse neuron types within a network, considering architecture design, optimization strategies, connectivity schemes, and the nuanced interplay with sparsity.

This revision offers a more thorough and nuanced exploration of the architectural considerations relevant to SFA, reflecting the intricate interactions present in biological networks and the potential benefits and problems in terms of information processing and network efficiency. By addressing these aspects, we believe our paper now provides a more insightful and complete overview of the architectural landscape in SFA research.

- First paragraph of page 11: hyperparameter tuning: Have you detected a relation between application examples of the covered literature and the architecture and parameters of the SNN used?

Thank you for pointing out this essential aspect of hyperparameter tuning in RSNN, specifically focusing on the adaptation time for threshold voltage and its relationship with input data timesteps (i.e., τ).

In our review of the literature, we observed that initial studies indicated a strong proportional relationship between tau and adaptation time for threshold voltage in RSNN. However, subsequent research has provided insights that challenge this relationship, showing that the problem can be solved even at lower τ values.

Interestingly, we noted a lack of standard benchmarks across the various studies to critically assess the relationship between hyperparameters. This absence of standardization highlights a significant gap in the field and potentially hinders the ability to draw definitive conclusions.

In our revised section on hyperparameter tuning, we have synthesized these findings to present a more comprehensive and nuanced view of the current understanding of hyperparameter significance in SFA models. We have also emphasized the need for more focused research in this area, including standardized approaches, to unravel the complexities and contribute to a deeper understanding.

“

The hyperparameter tuning of SFA models poses a complex problem, demanding an intricate balance between biological formalism and computational efficiency. The grid-based search

methods typically employed may fall short in such complex scenarios. An exploration of advanced optimization techniques, such as Bayesian optimization or gradient-based optimization, is suggested as a possible avenue for more intelligently and efficiently navigating the parameter space specific to SFA models. Systematic ablation studies could enhance this process by elucidating the effects of individual parameters and their interactions, potentially leading to a deeper understanding of hyperparameter significance.

”

- At the end of section VI.B you offer a list of possible applications where these models with SNN can be beneficial. This part should be expanded a bit with deeper definitions, clarifications, justifications, and discussions.

- The list of new research opportunities requires major review: a list is not adequate. Expand and discuss. Give your opinion based on collected information from literature and what you see could be the future.

Section VI has been thoroughly revised and updated in view of the neuron models with spike frequency adaptation. Section VI is now divided into two sub-sections A. Challenges and Roadmap, and B. Future opportunities. Sub Section A covers the following subtopics Encoding Techniques in Spike Frequency Adaptation (SFA), Learning Algorithms and Adaptive Neurons, Network Architecture and Connectivity, Hyperparameter Tuning and Mathematical Complexity of SFA Models, and Integration and Hardware Compatibility. Subsection B include Sustainable AI acceleration through Emerging NVM devices exploiting SFA, Real-Time Adaptation in Dynamic Environments, Continuous Learning and Temporal Feature Extraction, Enhanced Robustness Against Adversarial Attacks, and Regularization and Meta-Learning.

- The conclusion section is vague. Expand and provide concrete conclusions on your survey. Thank you for your comment. The conclusion has now been removed from the modified manuscript.

- Figure 5-a: why organic roadmap? What is the reason for using the word organic here? Thank you for your comment. Section VI has been restructured and rewritten in view of the adaptive neuron models. In view of the present discussion, Fig. 5-a in the modified manuscript is merged with Fig. 4.

- Title should be less generalistic since your review is not so broad in general terms (implementations for example)
Thank you for your comment. The title of the paper has now been changed to “Spike Frequency Adaptation: Bridging Neural Models and Neuromorphic Applications”

Reviewers' comments:

Reviewer #2 (Remarks to the Author):

I recommend acceptance.

Reviewer #3 (Remarks to the Author):

The revised manuscript has improved significantly. The authors' vision is clearer and the "Challenges and Roadmap" section helps the reader to understand the relevance of discussing spike frequency adaptation within neuron models. This being said, I still have the following concerns:

The thesis is still unclear in both the abstract and introduction. The authors must provide context for the reader as to why this paper is relevant for them. I suggest alluding to some of the thoughts that you describe in the roadmap. For example, a sentence such as the following would provide the reader more motivation:

"This paper contends that SFA is an integral feature which must be explored within neuron models moving forward. It has the potential to accelerate AI technologies, improve real-time adaptation of autonomous systems, as well as other benefits outlined in Section VI."

The citation of the Furber paper (2) is used to back up the claim that SNNs are computationally efficient but the Furber paper concludes the opposite, that ANNs are still more efficient until new ways of exploiting spiking neurons can be found.

The grammar still requires significant improvement.

Examples:

Currently:

"However, Integrate and Fire (IF) models [10] which mimic the activities of a biological neuron via functionalities of a simple Resistance (R)-Capacitance (C) electrical circuit is very popular due its simple elegant mathematical structure."

Corrected:

"However, Integrate and Fire (IF) models [10] which mimic the activities of a biological neuron via functionalities of a simple Resistance (R)-Capacitance (C) electrical circuit are very popular due to their simple elegant mathematical structure."

Currently:

"The remainder of the paper is organised as follows: Why to use an adaptive neuron models is presented in section II."

Corrected:

"The remainder of the paper is organised as follows: Reasons for using adaptive neuron models are presented in section II."

When an acronym is pronounced with a vowel sound, remember to use "an" instead of "a" in front of it.

For example, even though it is “a spiking neural network” it is “an SNN”.

Minor issues:

The end of the first paragraph in the Introduction does not flow well. I suggest the following:

“An enhanced version of the IF model is the Leaky Integrate and Fire (LIF) model that takes the membrane voltage leak into account. To further increase biological plausibility other models add spike frequency adaptation (SFA), i.e. increase in the inter-spike interval (ISI) over time for a regular spike train, which is an intrinsic feature of biological neurons. This paper focuses on SFA as an important feature to explore.”

In Section B. Advantages of spike frequency adaptation, you mention “artificial neural networks” at the beginning of the third paragraph. However, the source you cite uses spiking neurons, I believe this should be changed to “spiking neural networks”.

The list of simulators in section 5A is not comprehensive, this should be stated. You can also consider adding Neuron and Brian to the list.

I prefer to see another draft before I can provide a recommendation for acceptance or not.

Reviewer #4 (Remarks to the Author):

Thanks to the authors for taking careful care of my previous concerns. I think that the paper is now prepared for publication from my side.

Point by Point Response to Reviewers comments

Reviewers' comments:

Reviewer #2 (Remarks to the Author):

I recommend acceptance.

Response : Thank you for recommending acceptance.

Reviewer #3 (Remarks to the Author):

The revised manuscript has improved significantly. The authors' vision is clearer and the "Challenges and Roadmap" section helps the reader to understand the relevance of discussing spike frequency adaptation within neuron models. This being said, I still have the following concerns:

Response : Thank you for careful review of our manuscript. We tried to implement the changes suggested.

The thesis is still unclear in both the abstract and introduction. The authors must provide context for the reader as to why this paper is relevant for them. I suggest alluding to some of the thoughts that you describe in the roadmap. For example, as sentence such as the following would provide the reader more motivation:

"This paper contends that SFA is an integral feature which must be explored within neuron models moving forward. It has the potential to accelerate AI technologies, improve real-time adaptation of autonomous systems, as well as other benefits outlined in Section VI."

Response : We tried to update the abstract as follows

"The human brain's unparalleled efficiency in executing complex cognitive tasks has inspired the development of Spiking Neural Networks (SNNs). By harnessing neuron models equipped with spike frequency adaptation (SFA), these networks achieve computational enhancement while optimizing energy consumption. This review explores various adaptive neuron models from computational neuroscience, emphasizing their significance in artificial intelligence applications and hardware integration. It underscores the challenges and untapped potential of these models in spearheading energy-efficient neuromorphic systems."

The citation of the Furber paper (2) is used to back up the claim that SNNs are computationally efficient but the Furber paper concludes the opposite, that ANNs are still more efficient until new ways of exploiting spiking neurons can be found.

Response : Thank you for pointing out the misinterpretation of the Furber paper. We acknowledge this oversight and we have corrected the citation.

The grammar still requires significant improvement.

Examples:

Currently:

“However, Integrate and Fire (IF) models [10] which mimic the activities of a biological neuron via functionalities of a simple Resistance (R)-Capacitance (C) electrical circuit is very popular due its simple elegant mathematical structure.”

Corrected:

“However, Integrate and Fire (IF) models [10] which mimic the activities of a biological neuron via functionalities of a simple Resistance (R)-Capacitance (C) electrical circuit are very popular due to their simple elegant mathematical structure.”

Currently:

“The remainder of the paper is organised as follows: Why to use an adaptive neuron models is presented in section II.”

Corrected:

“The remainder of the paper is organised as follows: Reasons for using adaptive neuron models are presented in section II.”

When an acronym is pronounced with a vowel sound, remember to use “an” instead of “a” in front of it. For example, even though it is “a spiking neural network” it is “an SNN”.

Response : Thank you for your thorough review. We've made the suggested changes and carefully checked the grammar. We appreciate your guidance. We highlighted all the changes you suggested. Further, we took care grammar of whole paper.

Minor issues:

The end of the first paragraph in the Introduction does not flow well. I suggest the following:

“An enhanced version of the IF model is the Leaky Integrate and Fire (LIF) model that takes the membrane voltage leak into account. To further increase biological plausibility other models add spike frequency adaptation (SFA), i.e. increase in the inter-spike interval (ISI) over time for a regular spike train, which is an intrinsic feature of biological neurons. This paper focuses on SFA as an important feature to explore.”

In Section B. Advantages of spike frequency adaptation, you mention “artificial neural networks” at the beginning of the third paragraph. However, the source you cite uses spiking neurons, I believe this should be changed to “spiking neural networks”.

The list of simulators in section 5A is not comprehensive, this should be stated. You can also consider adding Neuron and Brian to the list.

Response : Thank you for the careful review. We tried to implement all the changes suggested and highlighted in the manuscript.

I prefer to see another draft before I can provide a recommendation for acceptance or not.

Reviewer #4 (Remarks to the Author):

Thanks to the authors for taking careful care of my previous concerns. I think that the paper is now prepared for publication from my side.

Response : Thank you for recommending acceptance.

REVIEWERS' COMMENTS:

Reviewer #3 (Remarks to the Author):

The manuscript has improved and I recommend acceptance.

A minor note, at the end of section 2.2 you write "When compared to LIF models, this reduction in spikes has the potential to decrease computational efficiency." I suppose you mean increase computational efficiency?

Also, please read through the manuscript again carefully for minor grammatical errors / spelling errors. A few examples I found:

Section 1

"However, Integrate and Fire (IF) models [12] which mimic the activities of a biological neuron via functionalities of a simple Resistance (R)-Capacitance (C) electrical circuit is very popular..."

Should be "...are very popular"

"For language processing short-term storage and integration of information in working memory in necessary."

Should be "...is necessary."

Section 2.2

SSNs are referred to, I expect you mean SNNs

There are also several places in the text where you capitalize "An" in the middle of a sentence.

Thank you for addressing my previous comments and congrats on the paper.